# Nanotwin architecture and ultra-high valley degeneracy lead to high thermoelectric performance in GeTe-based thermoelectric materials

Song Li[1,6], Yuxuan Yang [ID][2,6], Xiaoyu Fei[3,4], Yang Geng[1], Jiajun Nan[1], Pubao Peng[1], Guizhong Li[1], Yang Zhang[2,5], Xiaobing Liu [ID][3,4], Yongsheng Zhang [ID][3,4] ✉, Haijun Wu [ID][2] ✉ & Guodong Tang [ID][1] ✉

Here, we achieve a high peak $ZT$ of 2.5 as well as an exceptional average $ZT$ of 1.9 through nanotwin architecture and inducing ultra-high valley degeneracy. We find that nanotwins, ordered vacancy arrays and point defects serve as intense phonon scattering centers for enhancing wide-frequency phonon scattering, resulting in ultralow lattice thermal conductivity in GeTe. Interestingly, density-functional theory calculations reveal that $CuBiS_2$ alloying realizes refined valence band alignment in GeTe, generating an ultra-high valley degeneracy of 22. The dramatic enhancement of the Seebeck coefficient induced by the ultra-high valley degeneracy contributes to remarkably enhanced power factor over a very wide temperature range. The maximum power factor reaches as high as 49 µW cm$^{-1}$ K$^{-2}$. Consequently, a high peak $ZT$ as well as a large average $ZT$ are realized in GeTe without involving toxic elements. Importantly, the presence of nanotwins boundaries in GeTe effectively provides adequate barriers to block dislocation motion, leading to excellent hardness and compressive strength. Our finding provides a feasible pathway to design fascinating thermoelectric materials with high thermoelectric performance and mechanical properties.

Thermoelectric materials and devices can realize direct interconversion between electricity and waste heat based on charge carrier and phonon transport only, showing great potential in power generation and electronic cooling[1]. The energy conversion efficiency is primarily related to the dimensionless figure of merit ($ZT = S^2\sigma T/\kappa_T$), where $\sigma$, $S$, $\kappa_T$, and $T$ denote the electrical conductivity, Seebeck coefficient, total thermal conductivity and absolute temperature, respectively[2,3]. $\kappa_T$ comprises both electronic thermal conductivity $\kappa_e$ and lattice thermal conductivity $\kappa_L$, expressed as $\kappa_T = \kappa_e + \kappa_L$. $S^2\sigma$ can be expressed in terms of power factor ($PF = S^2\sigma$), which is commonly used to represent the electrical transport properties of the material. However, strong interdependence among thermoelectric parameters prevents us from maximizing the final $ZT$ and energy conversion efficiency[4,5].

[1]School of Materials Science and Engineering, Nanjing University of Science and Technology, Nanjing, China. [2]State Key Laboratory for Mechanical Behavior of Materials, Xi'an Jiaotong University, Xi'an, China. [3]Key Laboratory of Quantum Materials under Extreme Conditions in Shandong Province, School of Physics and Physical Engineering, Qufu Normal University, Qufu, China. [4]Laboratory of High Pressure Physics and Material Science (HPPMS), Advanced Research Institute of Multidisciplinary Sciences, Qufu Normal University, Qufu, China. [5]Electronic Materials Research Laboratory (Key Lab of Education Ministry), School of Electronic and Information Engineering and Instrumental Analysis Center, Xi'an Jiaotong University, Xi'an, China. [6]These authors contributed equally: Song Li, Yuxuan Yang. ✉e-mail: yshzhang@qfnu.edu.cn; wuhaijunnavy@xjtu.edu.cn; tangguodong@njust.edu.cn

PbTe, with a leading thermoelectric performance, has been used for power generation applications and space exploration missions[6,7]. Yet, the environmentally hazardous element Pb is creating pollution problems. GeTe has attracted extensive research owing to both its high thermoelectric performance and environmentally friendly features. GeTe undergoes a ferroelectric phase transition from the low-temperature rhombohedral phase (r-GeTe) to the high-temperature cubic phase (c-GeTe) at approximately 670 K, driven by the lone pair effect of Ge atoms, which induces Peierls distortion through their displacement along the [111] direction in two sublattices[8-10]. The improved lattice symmetry during phase transition will increase the valley-degeneracy ($N_v$) and provide an additional degree of freedom for optimizing the thermoelectric properties[11]. High intrinsic carrier concentration (~$10^{21}$ cm$^{-3}$) induced by the low forming energy of Ge vacancy results in much inferior thermoelectric performance in pristine GeTe[10,12]. Carrier concentration modulation via heterovalent doping by Bi[13] and Sb[14] doping has been demonstrated as an effective strategy to enhance the thermoelectric properties of GeTe, and increase formation energy of Ge vacancies can also decrease carrier concentration[15,16]. Band convergence, achieved through Cd[17], Zn[18], Ca[19], and Hg[20] doping in GeTe by reducing the energy offset between heavy and light bands, alongside resonance levels introduced by In[21] and Ga[22] doping that create a hump near the Fermi level in the density of states, both effectively enhance the Seebeck coefficient. Enhancing phonon scattering through lattice imperfections (e.g., point defects[23], vacancy clusters[24], dislocations[25], and planar vacancies[26]), secondary phases[27], and grain boundaries[28] serve as the primary strategy for reducing lattice thermal conductivity.

However, many of the above-mentioned strategies primarily focus on enhancing peak ZT, which constrains the energy conversion efficiency and practical application. Thermoelectric properties of GeTe systems have not yet reached the theoretical optimum[10,29]. Developing highly effective GeTe with not only high peak ZT but also high average ZT over a wide temperature range is a priority for achieving high conversion efficiency. Additionally, GeTe-based thermoelectric materials exhibit limited mechanical properties. It is a great challenge to simultaneously achieve highly competitive thermoelectric performance and good mechanical properties in thermoelectrics. Twin boundaries can act as intense phonon scattering center, improving thermoelectric properties by suppressing the lattice thermal conductivity[30]. In the meantime, twin boundaries also can lead to strengthening effect and enhanced mechanical performance[31]. Electronically, the construction of nanostructures inevitably induces carrier scattering, thereby degrading electrical properties. To overcome this drawback, band convergence of electron valence band edges can remarkably increase $N_v$ to provide additional carrier transport channels for optimizing electrical transport performance.

Here, a strategy of synergy of nanotwin architecture and inducing ultra-high valley degeneracy was proposed for simultaneously achieving high thermoelectric performance as well as excellent mechanical properties in GeTe (Fig. 1). Dense nanotwins combined with vacancy arrays and point defects significantly diminish lattice thermal conductivity by strengthening wide-frequency phonon scattering (Fig. 1a). In the meanwhile, an ultra-high $N_v$ of 22 was induced via refined valence band alignment between the three valence band maxima by CuBiS$_2$ alloying (Fig. 1b), resulting in largely enhanced power factor over the whole temperature range. Resultantly, these synergistic effects contributed to a peak ZT value of 2.5 and an average ZT of 1.9 across the temperature range of 400-823 K in GeTe (Fig. 1c). Furthermore, nanotwins boundaries contribute to excellent

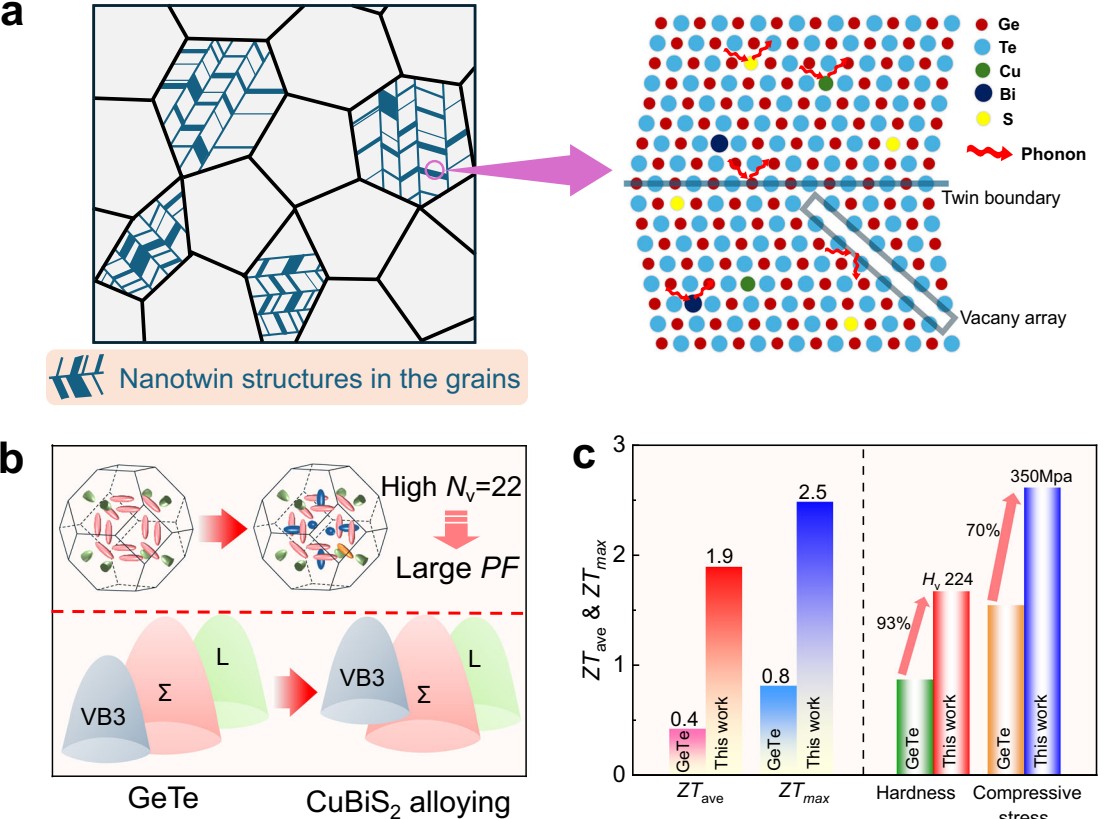

**Fig. 1 | Nanotwin architecture and ultra-high valley degeneracy leads to high thermoelectric performance and mechanical properties. a** Modulation mechanism of nanotwins, ordered vacancy arrays, and point defects on phonon transport. **b** CuBiS$_2$ alloying leads to band alignment and ultra-high valley degeneracy ($N_v$) of 22. **c** Comparison of thermoelectric performance and mechanical properties between pristine GeTe and (GeTe)$_{0.93}$(CuBiS$_2$)$_{0.07}$ sample.

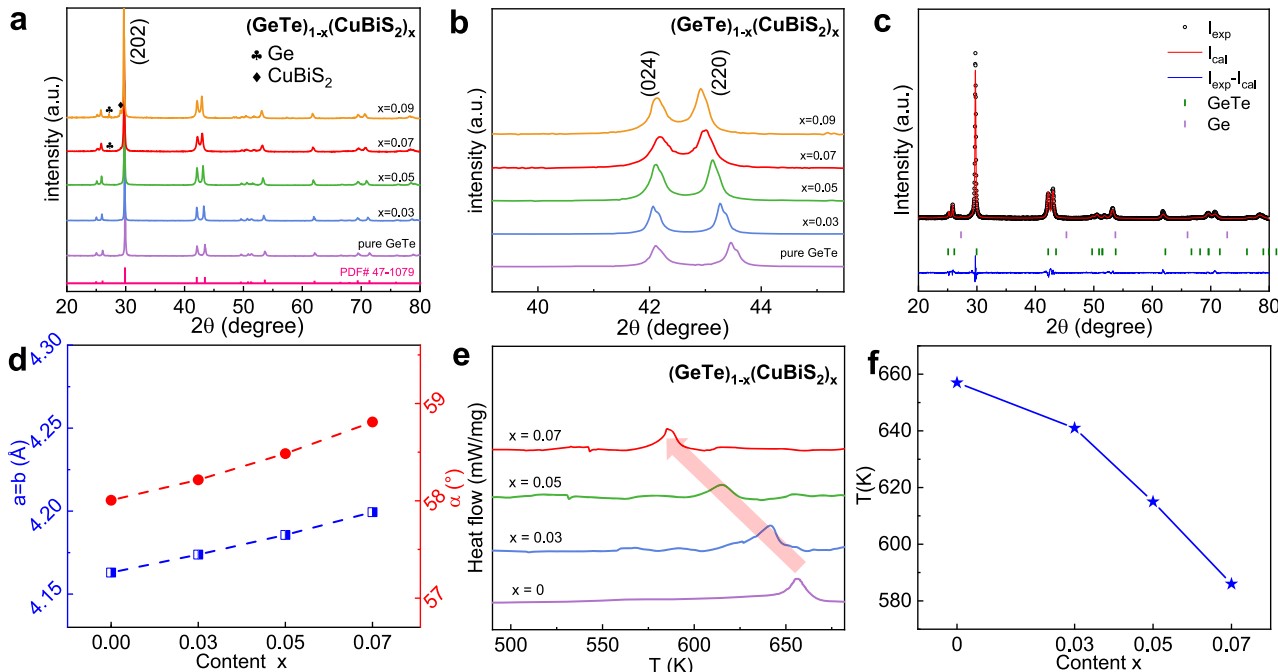

**Fig. 2 | The crystal structure and DSC characterization for (GeTe)$_{1-x}$(CuBiS$_2$)$_x$ samples. a** X-ray diffraction patterns of (GeTe)$_{1-x}$(CuBiS$_2$)$_x$ powders at room temperature after SPS. **b** The enlarged view of (024) and (220) diffraction patterns in the 2$\theta$ range of 40° ~ 46° from (**a**). **c** Rietveld refinement details of (GeTe)$_{0.93}$(CuBiS$_2$)$_{0.07}$ sample. **d** Lattice parameters extracted from XRD cell refinement. **e** Differential scanning calorimetry (DSC) analysis for (GeTe)$_{1-x}$(CuBiS$_2$)$_x$ samples. **f** Transition temperature ($T_p$) as a function of CuBiS$_2$ alloying content x.

mechanical properties in GeTe by hindering dislocation motion and accommodating dislocations. This work establishes a paradigm for high-efficiency and reliable thermoelectric materials design.

## Results and discussion
### Crystal structure and phase description
Powder X-ray diffraction (XRD) patterns of (GeTe)$_{1-x}$(CuBiS$_2$)$_x$ samples shown in Fig. 2a well match the rhombohedral GeTe structure (space group R3m)[32]. Extra peaks of Ge precipitates can be detected, which is an unavoidable phenomenon in GeTe system due to the low formation energy of Ge vacancies[32,33]. As the CuBiS$_2$ alloying content reaches x = 0.09, distinct diffraction peaks indexed to the secondary phase of CuBiS$_2$ were observed. This suggests that the solubility limit of CuBiS$_2$ in GeTe is lower than 9%. On the other hand, the position of (024) and (220) peaks shift toward lower angles and gradually merge with the increase of CuBiS$_2$ (Fig. 2b), implying the expansion of lattice and increase of lattice symmetry[34]. Rietveld refined XRD patterns and the refinement details can be found in Fig. 2c, d and Supplementary Fig. 1, respectively. Figure 2d clearly show that the lattice constant a and b increase from 4.16 Å to 4.20 Å with increasing CuBiS$_2$ alloying content, which can be attributed to the fact that the ionic radii of Bi$^{3+}$ (1.03 Å) and Cu$^+$ (0.77 Å) are both larger than that of Ge$^{2+}$ (0.73 Å). The occupied positions of the corrected atoms are exhibited in Supplementary Table 1. It is found that the Cu and Bi atoms occupy the Ge site (Wyckoff position 2c) with coordinates of (0, 0, 0.2437), while the S atoms occupy the Te site (Wyckoff position 2c) with coordinates of (0, 0, 0.7629). Furthermore, the phase constitution and quantitative phase fractions present in the alloyed samples are clarified (Supplementary Table 2). It proves the good dissolution of CuBiS$_2$ in GeTe when the alloying content is less than or equal to x = 0.07. Because the phase transition progress from *r*-GeTe to *c*-GeTe is caused by center cation atom shifting along the [111] direction (Peierls distortion), we can use the lattice angle $\alpha$ to represent the lattice symmetry[10]. The lattice angle $\alpha$ enlarges from 58° to 58.8°, confirming the enhanced

lattice symmetry induced by CuBiS$_2$ alloying. The differential scanning calorimetry (DSC) investigations in Fig. 2e were performed to investigate the phase transition temperature of (GeTe)$_{1-x}$(CuBiS$_2$)$_x$ samples. The phase transition temperature ($T_p$) decreases with the increase of CuBiS$_2$ content (Fig. 2f). In particular, the phase transition temperature gradually shifted down to ~ 586 K with x = 0.07, which is about 70 K lower than the pure GeTe. The reduced phase transition temperature further indicates that the lattice symmetry of GeTe has increased, which extends the superior thermoelectric transport behavior of cubic phase to low temperatures[21,35]. The fracture morphology suggests the sample is very compact with no pores (Supplementary Fig. 2). The relative density of all the samples exceeds 96% (Supplementary Table 3), confirming the highly compact nature of the sample.

### Thermal transport properties
The temperature-dependent total thermal conductivity ($\kappa_T$) experienced a sharp decrease after CuBiS$_2$ alloying across the whole temperature range compared to the pure GeTe (Fig. 3a). To determine the factors responsible for the variation in $\kappa_T$, the temperature-dependent electrical thermal conductivity ($\kappa_e$) is calculated according to the equation of Wiedemann-Franz law ($\kappa_e = L\sigma T$). The Lorenz number ($L$) is calculated by fitting respective Seebeck coefficient values with an assumption of a single parabolic band (SPB) model (Supporting Information), as illustrated in Supplementary Fig. 3. $\kappa_e$ was greatly suppressed because of the reduced $\sigma$, which is partially responsible for the reduced $\kappa_T$. The rising trend of $\kappa_e$ at high temperature is caused by electron thermal excitation. By subtracting $\kappa_e$ from the $\kappa_T$, we achieved the lattice thermal conductivity ($\kappa_L$) in Fig. 3b. It is demonstrated that the $\kappa_L$ of all the CuBiS$_2$ alloying samples is effectively suppressed in the whole temperature range and decreases with the rising alloying content. However, an anomalous increase overshadows the decreasing trend at high temperatures, indicating bipolar thermal-conduction activation. The lowest $\kappa_L$ of best performance (GeTe)$_{0.93}$(CuBiS$_2$)$_{0.07}$ sample is restrained to ~0.38 W m$^{-1}$K$^{-1}$ at 673 K, a value close to the

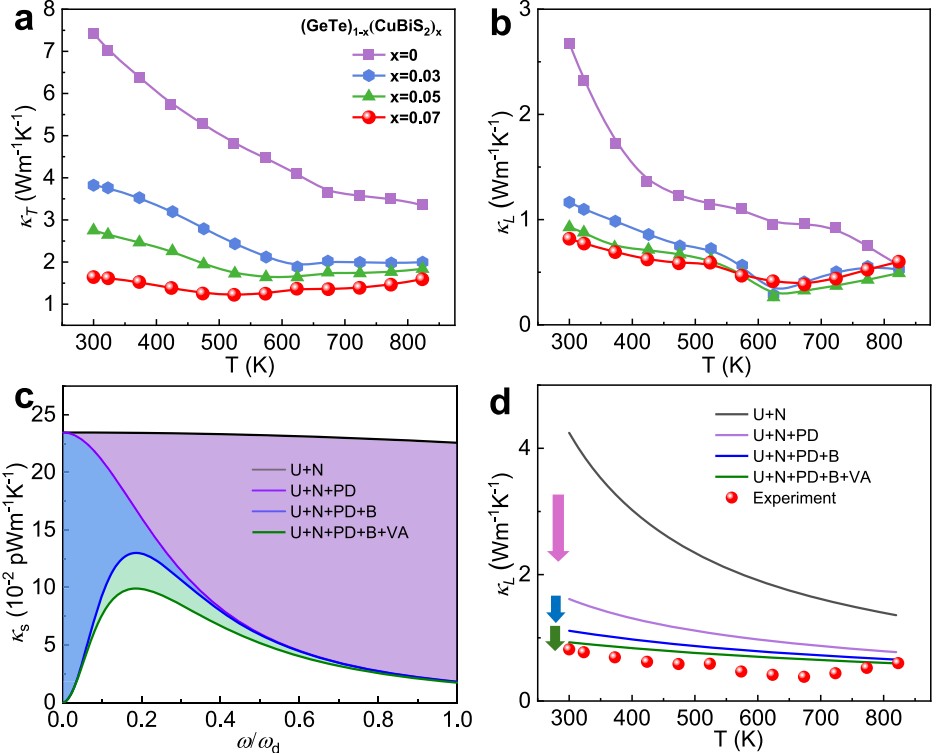

**Fig. 3 | Phonon transport properties and mechanism of $(GeTe)_{1-x}(CuBiS_2)_x$ samples. a** Total thermal conductivity ($\kappa_T$). **b** Lattice thermal conductivity ($\kappa_L$). **c** Calculated spectra lattice thermal conductivity ($\kappa_s$) by the Debye-Callaway model, with different phonon scattering mechanisms at 300 K. **d** Temperature dependent

the calculated $\kappa_L$ by the Debye-Callaway model includes different scattering mechanisms and is compared with the experimental values of the $(GeTe)_{0.93}(CuBiS_2)_{0.07}$ sample.

Cahill model theoretical limit (~0.3 W m$^{-1}$ K$^{-1}$) of GeTe[36], which is lower than most recent reported GeTe systems (Supplementary Fig. 4)[37–43].

## Microstructural characterization

To elucidate the underlying mechanism of markedly reduced $\kappa_L$, microstructural characterization was carried out utilizing scanning transmission electron microscopy (STEM). In the high-angle annular dark-field (STEM HAADF) images of Supplementary Fig. 5, the strip ferroelectric domains are found in the matrix. In the meantime, abundant nanotwins can be found in the matrix, as shown in Fig. 4a. The magnified image (Fig. 4b) clearly shows the domain wall between the twin structures. The atomic-level (high-resolution transmission electron microscopy, HRTEM) image of nanotwins is shown in Fig. 4c. It is clear that the atomic arrangements exhibit mirror symmetry across the twin boundaries, which is the typical structure of nanotwins. FFT (fast Fourier transformation) pattern of Fig. 4c for nanotwins, spot splitting away from the transmitted beam (T) was observed (Fig. 4d). This further confirms the twin structures. The intensity in the STEM HAADF image is approximately proportional to the atomic number $Z^2$. Z-contrast intensity profiles of Te and Ge atomic columns taken from Fig. 4c with corresponding histograms are presented in Fig. 4e, i and f, and j. The reddish hues correspond to heavier elements, while bluish tones indicate lighter elements and vacancies. The intensity distribution of Te columns is more uniform than that of Ge columns, the Z-contrast intensity of the Ge atomic columns exhibits significant local variations (inhomogeneity), indicating pronounced compositional fluctuations at the microscopic scale. Such local compositional fluctuations serve as a key microstructural mechanism for introducing mass-field perturbations and enhancing phonon scattering. Additionally, due to the low formation energy of Ge vacancies, such vacancies likely present in the matrix also appear as blue signals.

The random substitution of atoms with varying masses within the GeTe lattice introduces local mass fluctuations, which enhance phonon scattering and significantly reduce thermal conductivity. Bond length calculations for both Te and Ge sites are shown in Fig. 4g, k and h, l. Reddish lines denote longer bonds, bluish lines represent shorter bonds, and green lines indicate the average bond lengths. The Te−Te and Ge−Ge bond lengths fluctuate in the range of 94−107% and 94−109% of their average values, respectively. This bond-length variation is attributed to local lattice strain induced by nanotwins, which results in a more flexible lattice framework. The softened lattice promotes large-amplitude atomic vibrations, significantly enhancing anharmonic effects and thereby strengthening phonon scattering, which leads to a notable reduction in thermal conductivity.

Ordered vacancy arrays can be observed in the HRTEM image in Fig. 5a, and the corresponding (geometric phase analysis) GPA map is exhibited in Fig. 5b−d. GPA analysis indicates a large average lattice strain fluctuation, suggesting the presence of localized lattice strain at the vacancy arrays. Atomic-resolution HAADF image of the ordered vacant arrays is illustrated in Fig. 5e. Along this zone axis, Ge and Te atomic columns exhibit distinct contrast: Ge columns appear smaller and dimmer due to lower atomic number, while Te columns display larger, brighter features. The corresponding intensity line profile along the yellow frame from the atom arrays in Fig. 5f also confirms the ordered vacant arrays. The schematic of the atom array of vacancy arrays and the complete lattice was shown in Fig. 5g to better understand the difference between them. Z-contrast intensity profiles and bond length calculations for Te and Ge atomic columns are presented in Fig. 5h−k. Flanking the ordered Ge-vacancy chains, a decrease in Te-column intensity alongside an increase in Ge-column intensity indicates the substitution of Te sites by lighter atoms and Ge sites by heavier atoms, facilitating the formation of one-dimensional ordered

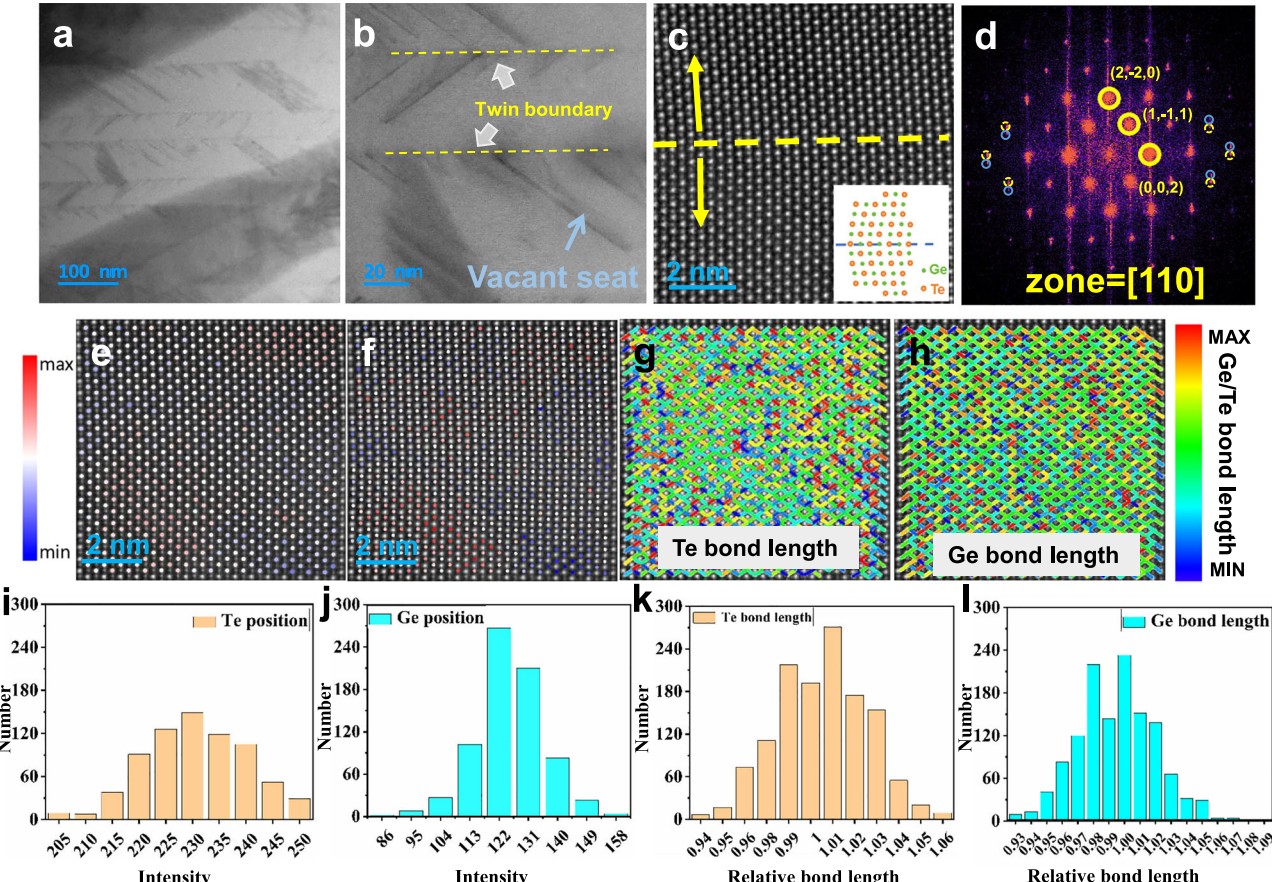

**Fig. 4 | Microstructure investigations of $(GeTe)_{0.93}(CuBiS_2)_{0.07}$ sample for twin structures. a** STEM image of abundant twin structures in the matrix. **b** HRTEM image of a showing the twin structure and domain wall. **c** Atomically-resolved STEM HAADF image of the twin structure along the [100] zone axis, with an inset schematic projection of the twin structure. **d** The FFT pattern from marked zone in (**c**). **e** Utilized the Z-contrast feature of STEM, drawing the Te-site intensity map. **f** Utilized the Z-contrast feature of STEM to draw the Ge-site intensity map. **g** Lattice spacing mapping for Te-site species, characterizing local lattice distortions associated with Te-Te bonds caused by doping. **h** Lattice spacing mapping for Ge-site species, characterizing local lattice distortions associated with Ge-Ge bonds caused by doping and vacancies. **i** Calculation of the Z-contrast of Te-site. **j** Calculation of the Z-contrast of Ge-site. **k** Calculation of the bond length of Te-site. **l** Calculation of the bond length of Ge-site.

Ge-vacancy chains. The ordered vacancy arrays introduce localized mass fluctuations and lattice strain, leading to lattice softening and a reduction in the lattice thermal conductivity.

Overall, the enhanced point defects lead to strong phonon scattering, thereby reducing the lattice thermal conductivity. Abundant nanotwins significantly enhance anharmonic effects and thereby strengthening phonon scattering through lattice softening, leading to a notable reduction in thermal conductivity. Furthermore, the ordered vacancy arrays significantly suppress the lattice thermal conductivity by introducing localized mass fluctuations and lattice strain. As a result, a remarkably low lattice thermal conductivity ($0.38\,W\,m^{-1}K^{-1}$) is achieved in $(GeTe)_{0.93}(CuBiS_2)_{0.07}$, benefiting from a constructed defect complex including point defects, nanotwins, and ordered vacancy arrays in GeTe.

To visualize the effects of various scattering center on the decreased $\kappa_L$, the frequency dependent spectra lattice thermal conductivity ($\kappa_s$) for $(GeTe)_{0.93}(CuBiS_2)_{0.07}$ sample is simulated by Debye-Callaway model (Supporting Information)[44], as shown in Fig. 3c, considering the contribution of Umklapp scattering process (U), normal scattering process (N), point defect scattering process (PD), grain boundaries and twin boundaries scattering process (B), vacant arrays scattering process (VA) (the detail parameters are listed in Supplementary Table 4). It is clearly shown that the point defects can largely scatter the medium and high-frequency phonons. Twin boundaries and vacancy arrays are another important mechanism to scatter low

and medium-frequency phonons. The theoretical $\kappa_L$ is calculated and compared with the experimental value in Fig. 3d. The calculated results confirm that nanotwins, ordered vacancy arrays, and point defects indeed significantly reduce the lattice thermal conductivity of GeTe. The defect complex, including nanotwins, ordered vacancy arrays, and point defects, serves as intense phonon scattering centers for enhancing wide-frequency phonon scattering (Fig. 1a), thus contributing to the greatly suppressed $\kappa_L$ for the $(GeTe)_{0.93}(CuBiS_2)_{0.07}$ sample.

### Electrical transport properties

The temperature-dependent electrical conductivity ($\sigma$) of $(GeTe)_{1-x}(CuBiS_2)_x$ samples is shown in Fig. 6a. $\sigma$ decreases with increasing $CuBiS_2$ alloying content. $(GeTe)_{0.93}(CuBiS_2)_{0.07}$ material exhibits lowest $\sigma$ in the whole temperature range. The sudden change of $\sigma$ from 550 to 670 K can be attributed to the valance band switch between L and $\Sigma$ points induced by phase transition[27]. Hall measurements were performed to determine the carrier concentration ($n$) and carrier mobility ($\mu$), as illustrated in Fig. 6c. The carrier concentration decreases with the increasing content of $CuBiS_2$. The $n$ decreases from $6.5 \times 10^{20}\,cm^{-3}$ for pure GeTe to $2.28 \times 10^{20}\,cm^{-3}$ for $(GeTe)_{0.93}(CuBiS_2)_{0.07}$ sample, which is suppressed to the optimal region for GeTe system. Bi serves as a donor dopant, effectively providing a large number of extra electrons, which can compensate for the intrinsic high hole concentration in GeTe[45]. In the meanwhile, the introduction of Cu is an efficient dopant for decreasing hole

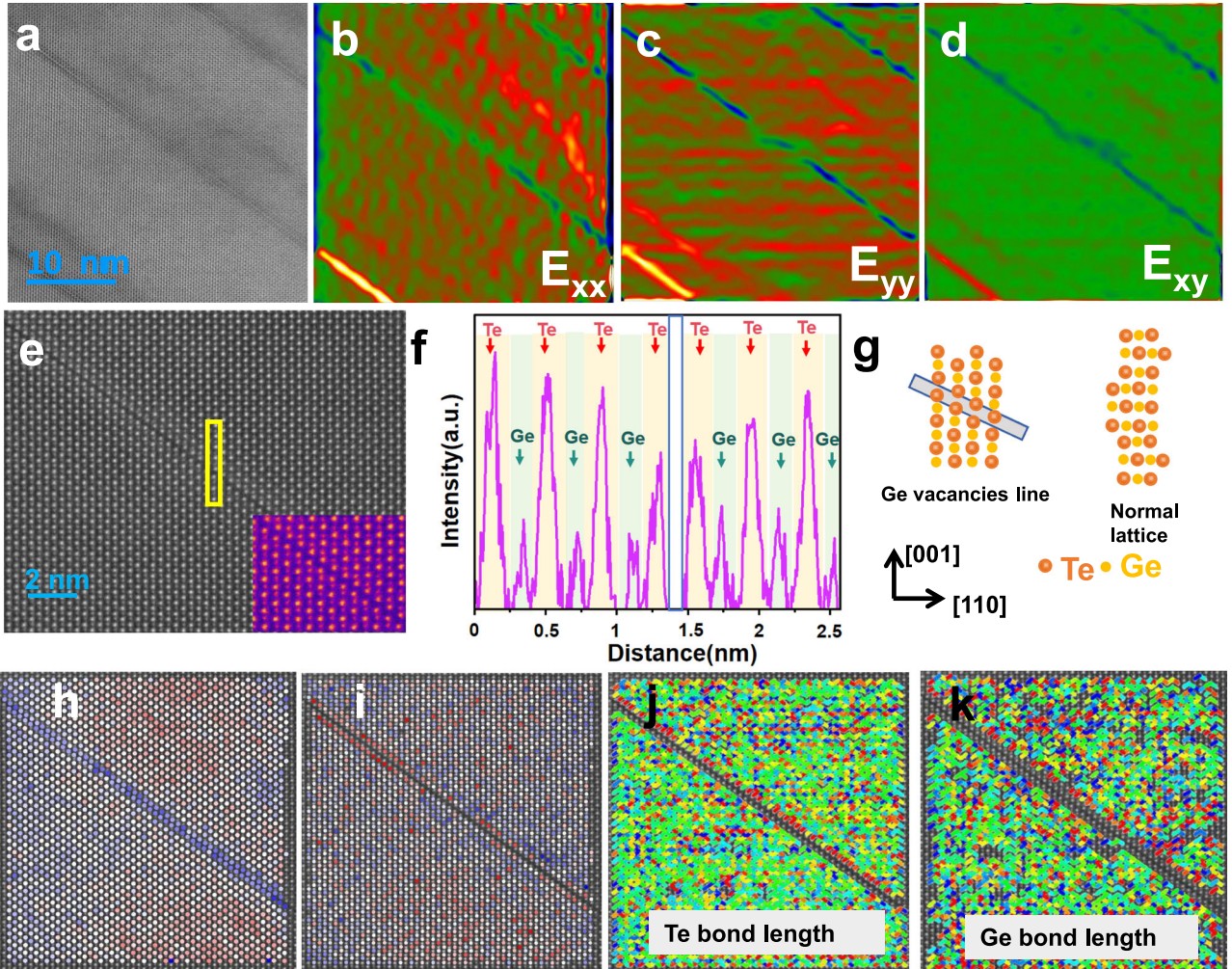

**Fig. 5 | Microstructure investigations of (GeTe)$_{0.93}$(CuBiS$_2$)$_{0.07}$ sample for vacant arrays. a** STEM image of abundant vacant arrays in the matrix. **b–d** GPA (geometric phase analysis) mapping of Figure (**a**). **e** Atomically-resolved STEM HAADF image of the vacant array. **f** Corresponding intensity line profile along the yellow frame from the atom arrays marked in (**e**). **g** Schematic projection of the vacant array and normal lattice. **h** Utilized the Z-contrast feature of STEM, drawing the Te-site intensity map. **I** utilized the Z-contrast feature of STEM, drawing the Ge-site intensity map. **j** Lattice spacing mapping for Te-site species, characterizing local lattice distortions associated with Te-Te bonds caused by vacant array. **k** Lattice spacing mapping for Ge-site species, characterizing local lattice distortions associated with Ge-Ge bonds caused by vacant array.

concentration by increasing the vacancy formation energy[46]. $\mu$ experienced a sharp decrease after CuBiS$_2$ alloying due to enhanced point defect scattering. The reduced carrier concentration and carrier mobility lead to the decline in $\sigma$.

The temperature dependence of the Seebeck coefficient ($S$) for (GeTe)$_{1-x}$(CuBiS$_2$)$_x$ samples (Fig. 6b) exhibits positive values, indicating p-type conductivity consistent with Hall measurements. It is found that all CuBiS$_2$ alloyed samples exhibit significantly enhanced $S$ among the whole temperature range. $S$ at 823 K increases from -133 μV K$^{-1}$ for the pure GeTe to -240 μV K$^{-1}$ for the high performance (GeTe)$_{0.93}$(CuBiS$_2$)$_{0.07}$ sample, exhibiting great advantages compared with other reported state-of-the-art GeTe based systems (Supplementary Fig. 6)[17,29,32,47–49]. In order to elucidate the mechanism of large increase of the Seebeck coefficient of GeTe, the Pisarenko relation between $S$ and the carrier concentration ($n$) is calculated based on the single parabolic band (SPB) model. The density-of-state (DOS) effective mass $m^*$ is fitted by SPB model in Supplementary Table 5. All the CuBiS$_2$ alloying samples lie above the $m^* = 1.6$ $m_0$ line, and the $m^*$ improves with the increasing alloying content of CuBiS$_2$ from 1.03 $m_0$ for pure GeTe to 2.23 $m_0$ for (GeTe)$_{0.93}$(CuBiS$_2$)$_{0.07}$ sample. Previously

reported data of GeTe-CuSbSe$_2$[28], GeTe-CuBiSe$_2$[29], and GeTe-NaSbTe$_2$[31] are also included for comparison in Fig. 6d. The (GeTe)$_{1-x}$(CuBiS$_2$)$_x$ samples exhibit higher $m^*$ as compared to these alloying systems. The large enhancement of $m^*$ clearly predicts that CuBiS$_2$ alloying modifies the electronic band structures of GeTe.

To decode the mechanism of higher $m^*$ of (GeTe)$_{1-x}$(CuBiS$_2$)$_x$ alloying systems and understand the effects of CuBiS$_2$ alloying on the band structures of GeTe, we carrier out Density-functional theory (DFT) calculation of pure GeTe and (GeTe)$_{0.93}$(CuBiS$_2$)$_{0.07}$ in the cubic phase. The supercell used for calculating is shown in Supplementary Fig. 7. The $k$-point paths in the first Brillouin zone used for band structure calculations are shown in Supplementary Fig. 8. For the pristine GeTe compound (Fig. 6g), it exhibits the direct band gap ($E_g = 0.35$ eV) semiconductor with the valence band maximum (VBM or VBM1 in Fig. 6g) and conduction band minimum (CBM) at the L point in the Brillouin zone (BZ). The second and third valence band maxima (VBM2 and VBM3) are located at the Σ point and along the Γ-X direction, respectively. The valley degeneracy ($N_V$) of VBM1 (at the L point), VBM2 (at the Σ point), and VBM3 (along the Γ-X direction) are 4, 12, and 6, respectively. For the band offsets of these valence band maxima,

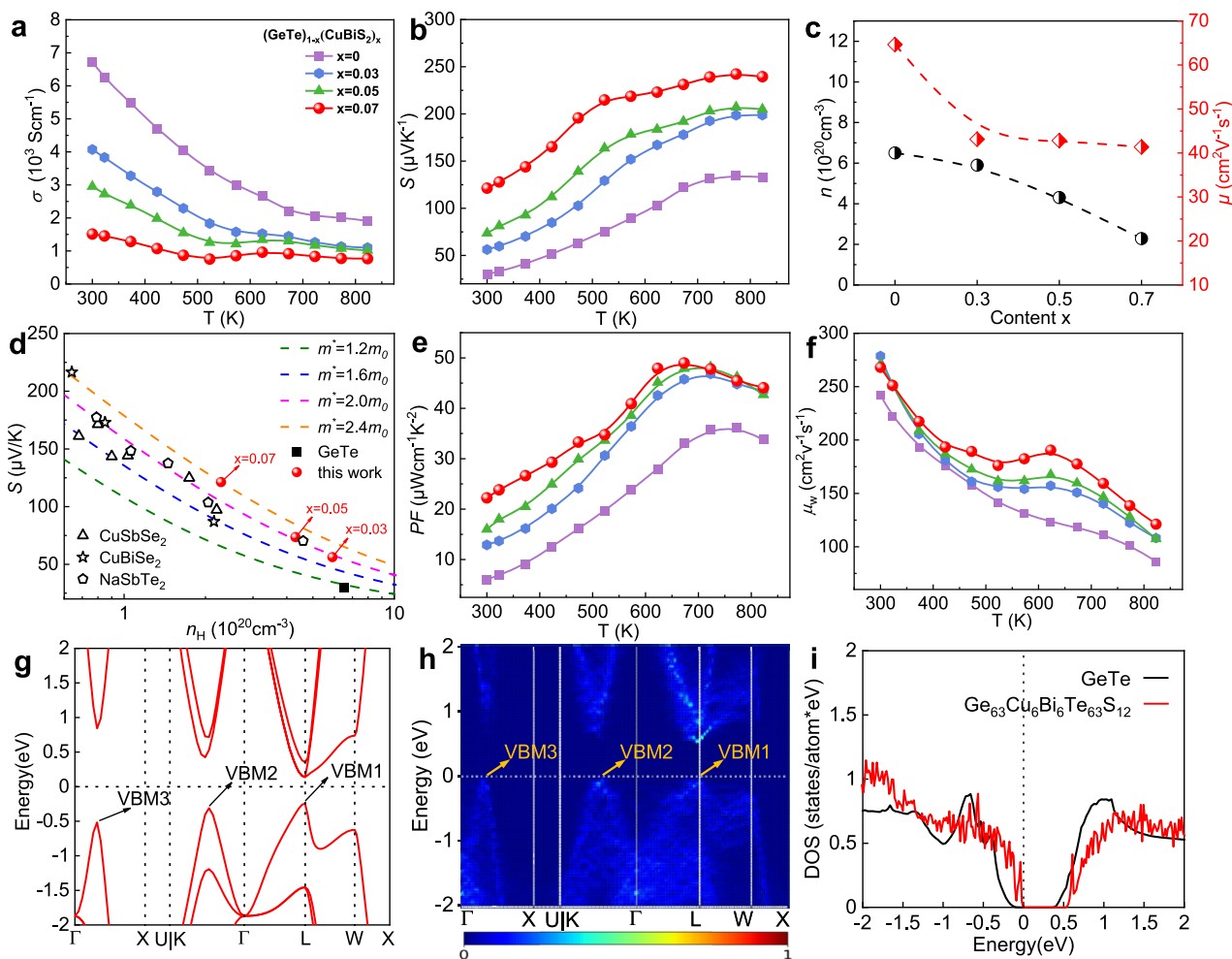

**Fig. 6 | Electronic transport properties and band structures of (GeTe)$_{1-x}$(CuBiS$_2$)$_x$ samples. a** Electrical conductivities ($\sigma$). **b** Seebeck coefficients ($S$). **c** Compositional dependence of carrier concentration ($n$) and carrier mobility ($\mu$) at room temperature. **d** The carrier concentration-dependent Seebeck coefficient at room temperature. **e** Power factor ($PF$). **f** The weight mobility ($\mu_w$). **g** Electronic band structures of cubic GeTe. **h** Electronic band structures of Ge$_{63}$Cu$_6$Bi$_6$Te$_{63}$S$_{12}$. The color bar represents the magnitude of the spectral weight, which characterizes the probability of the primitive cell eigenstates contributing to a particular supercell eigenstate of the same energy. **i** The density of states of GeTe (the black line) and Ge$_{63}$Cu$_6$Bi$_6$Te$_{63}$S$_{12}$ (the red line).

which are related to the Seebeck coefficients of samples, we notice that the energy difference between the VBM1 and VBM2 ($\Delta E^{VBM1\text{-}VBM2}$) is 67 meV. However, $\Delta E^{VBM1\text{-}VBM3}$ is quite large, 273 meV. Such large band offsets can hardly be recognized as the band convergency. Once the CuBiS$_2$ is alloyed in GeTe (Ge$_{63}$Cu$_6$Bi$_6$Te$_{63}$S$_{12}$), the electronic band structures are significantly changed (Fig. 6h). Interestingly, under the doping, the compound is changed to an indirect band gap semiconductor: the valence band maximum (VBM) is shift from the L point (the VBM1 position) to the Σ point (the VBM2 position) and the band gap is slightly enlarged to 0.49 eV. Obviously, by shifting the VBM from the L point to the Σ point, the $N_v$ is significantly increased from 4 to 12. Additionally, CuBiS$_2$ alloying clearly lifts the energy positions of VBM2 and VBM3, decreasing the band offset with respect to the VBM1 (Fig. 6h and Table 1): $\Delta E^{VBM1\text{-}VBM2} = -16$ meV and $\Delta E^{VBM1\text{-}VBM3} = 63$ meV (the negative band offset value of $\Delta E^{VBM1\text{-}VBM2}$ is due to the shift of VBM from VBM1 to VBM2). Thus, these small band offsets can be considered as the band alignment of VBM1, VBM2 and VBM3. Correspondingly, $N_v$ is increased to an ultra-high value of 4($N_v^{VBM1}$)+12($N_v^{VBM2}$)+6($N_v^{VBM3}$)=22. In previous reported works, they focused on promoting valley degeneracy of L and Σ. Cd[17], Ti[50], Ca[19], and Mn[51] elements doping in GeTe are proved to realize the band convergence between L and Σ. However, the energy difference between light and heavy bands was not effectively eliminated, weakening the contribution of heavy bands. Thus, the

**Table. 1 | The band offsets in GeTe and Ge$_{63}$Cu$_6$Bi$_6$Te$_{63}$S$_{12}$ among several valence band maxima, including VBM1 at the L point, VBM2 along the Γ-K direction and VBM3 along the Γ-X direction in Supplementary Fig. 8**

| Band offset (meV) | $\Delta E^{VBM1\text{-}VBM2}$ | $\Delta E^{VBM1\text{-}VBM3}$ |
|---|---|---|
| GeTe | 67 | 273 |
| Ge$_{63}$Cu$_6$Bi$_6$Te$_{63}$S$_{12}$ | −16 | 63 |

The negative band offset value of $\Delta E^{VBM1\text{-}VBM2}$ in Ge$_{63}$Cu$_6$Bi$_6$Te$_{63}$S$_{12}$ is due to the shift of VBM from VBM1 to VBM2.

actual contributing valley degeneracy ranges from 4 to 16. Then, a refined band alignment was realized in Zn-doped Ge$_{1-x}$Sb$_x$Te and V-doped Ge$_{1-x}$Bi$_x$Te samples[18,42], band edges between Σ and L nearly locate at the same energy level ($\Delta E^{VBM1\text{-}VBM2} \leq 0.01$ eV), leading to a total valley degeneracy of 16 ($N_v^{VBM1} = 4$, $N_v^{VBM2} = 12$). However, from the band structures of $c$-GeTe, we notice that there has the third valence band (VBM3, $N_v = 6$) along the Γ-X direction, which is always ignored in former works. If we could reduce the energy difference between third valence band and VBM1, we can further increase the $N_v$ and involve more channels for carrier transport. Therefore, by alloying CuBiS$_2$ in GeTe, we enable the band alignment not only between the VBM1 and VBM2, but also between VBM1 and VBM3, which contribute to a total

valley degeneracy of 22 ($N_v^{VBM1} = 4$, $N_v^{VBM2} = 12$, $N_v^{VBM3} = 6$). This is a much higher value than those of previous reported literature[8,17–19,42,51] (Supplementary Table 6). Therefore, the CuBiS$_2$ alloying induces VBM off the high-symmetry point in the BZ and the band alignment among the three valence band maxima, owing to the combined effect of CuBiS$_2$ alloying[43,52]. Both of them lead to the large $N_v$ and boost the electronic density of states (DOS) around the valence band maximum, which is clearly seen in Fig. 6i. The boosted DOS will induce the high DOS effective mass ($m^*$) and the correspondingly large Seebeck coefficient. Moreover, increasing $N_v$ is an effective way to enhance $m^*$ but avoiding deteriorating $\mu$[53], which is in good agreement with the experimental measurements.

CuBiS$_2$ alloying contributes to a sharp increase of power factor (PF) throughout the measured temperature range, as presented in Fig. 6e. The room temperature PF increases from 6 μW cm$^{-1}$ K$^{-2}$ for the pure GeTe to 22 μW cm$^{-1}$ K$^{-2}$ for (GeTe)$_{0.93}$(CuBiS$_2$)$_{0.07}$ sample. It is worth noting that a remarkable PF of ~49 μW cm$^{-1}$ K$^{-2}$ is achieved in the (GeTe)$_{0.93}$(CuBiS$_2$)$_{0.07}$ sample at 673 K due to markedly improved S. The power factor shows a significant degradation (Supplementary Fig. 9) as CuBiS$_2$ content further increases to x = 0.09. As CuBiS$_2$ alloying content exceeds the solid solubility limit, CuBiS$_2$ precipitates presents within the matrix. These precipitates cause strong carrier scattering, resulting in a sharp deterioration of electrical transport performance. The comparison of the PF for (GeTe)$_{0.93}$(CuBiS$_2$)$_{0.07}$ sample with other reported GeTe materials is shown in Supplementary Fig. 10[29,41,54–57]. The large integral area of PF demonstrates that CuBiS$_2$ alloying enhances electrical transport properties of GeTe over the whole temperature range, making it promising to achieve high wide-temperature-range thermoelectric performance. The weighted mobility ($\mu_w$) in Fig. 6f provides a direct assessment of the intrinsic electrical transport characteristics. Details of the calculation can be found in Supporting Information. $\mu_w$ exhibits a declining trend with rising temperature, which can be attributed to intensified carrier scattering caused by impurities. Notably, the (GeTe)$_{0.93}$(CuBiS$_2$)$_{0.07}$ sample possess significantly higher $\mu_w$ than pristine GeTe and demonstrates the highest $\mu_w$ among investigated samples, indicating its superior electrical transport characteristics compared to its counterparts.

### ZT values, energy conversion efficiency of the thermoelectric device, and mechanical properties of (GeTe)$_{1-x}$(CuBiS$_2$)$_x$ samples

ZT value of (GeTe)$_{1-x}$(CuBiS$_2$)$_x$ samples shows pronounced enhancement compared to pristine GeTe (Fig. 7a). An extraordinary peak ZT of ~2.5 at 723 K is achieved in (GeTe)$_{0.93}$(CuBiS$_2$)$_{0.07}$, which benefits from the synergistic optimization of thermal and electrical performance enabled by CuBiS$_2$ alloying. This high thermoelectric performance surpasses most of promising reported GeTe based thermoelectric materials (Supplementary Fig. 11)[29,38,58–62]. Repeated measurements in Supplementary Fig. 12 demonstrate good experimental reproducibility for such high performance. Moreover, the material maintains its thermoelectric performance after heating-cooling cycles, demonstrating excellent thermal stability (Supplementary Fig. 13). XRD measurements (Supplementary Fig. 14) reveal that the sample maintained consistent phase components before and after cycling thermoelectric measurements. Thermogravimetric analysis (TGA) results (Supplementary Fig. 15) reveal negligible weight loss after heating, confirming that elemental volatilization is negligible during heating. SEM and EDS analysis proves that the sample did not significantly degrade under high temperatures (Supplementary Fig. 16).

Attaining a high average ZT across a broad temperature range is greatly desirable for promoting the practical application of GeTe because thermoelectric devices need to operate over the several-hundred-kelvin operating range. It is worth noting that ZT is significantly enhanced in the whole temperature range (Supplementary Fig. 17) due to significantly enhanced PF and sharply reduced lattice thermal conductivity[22,29,38,55,58,60,63]. A record-high $ZT_{ave}$ (average ZT) of

1.9 was realized between 400 and 823 K. The wide-temperature-range thermoelectric performance outperforms most state-of-the-art p-type thermoelectric materials[64–68], enabling this GeTe-based alloy as one of the best medium-temperature thermoelectric materials. The $ZT_{ave}$ value of the (GeTe)$_{0.93}$(CuBiS$_2$)$_{0.07}$ sample is 442% higher than that of pure GeTe and outperforms utmost GeTe-based systems, as shown in Fig. 7b. A (GeTe)$_{0.93}$(CuBiS$_2$)$_{0.07}$ single leg device was fabricated to demonstrate the application potential. The output voltage (V), output power (P), and energy conversion efficiency ($\eta$) as functions of electric current (I) are shown in Fig. 7c, d and Supplementary Fig. 18. Experimentally, we achieved a high output power of 54 mW and a corresponding $\eta$ of ~12% under a temperature difference ($\Delta T$) of 470 K. The maximum efficiency outperforms most of the reported high-performance thermoelectrics, including those of SnSe[69], PbTe[70], Half-Heusler[71], Bi$_{0.4}$Sb$_{1.6}$Te$_{3.2}$[72], PbSe[73] and GeTe systems[29,40,74,75] (Supplementary Fig. 19). Except for superior thermoelectric performance, robust mechanical properties of GeTe materials are also important for practical applications. Vickers indentation method was used to measure the micro-Vickers hardness of (GeTe)$_{1-x}$(CuBiS$_2$)$_x$ samples. The hardness of the samples keeps rising with the increasing alloying content of CuBiS$_2$ in Fig. 7e. Especially, the best performance (GeTe)$_{0.93}$(CuBiS$_2$)$_{0.07}$ sample achieved a hardness of ~224 $H_v$, which is ~93% higher than that of the pure GeTe (~116 $H_v$). This result also exhibits great advantages compared to the other reported works[45,61,76]. Moreover, the compressive strain-stress of the (GeTe)$_{0.93}$(CuBiS$_2$)$_{0.07}$ sample is tested and shown in Fig. 7f. (GeTe)$_{0.93}$(CuBiS$_2$)$_{0.07}$ sample exhibits enhanced compressive strengths of 349 MPa, which is much higher than utmost reported GeTe thermoelectrics[77–79]. It is worth noting that the compressive strength of (GeTe)$_{0.93}$(CuBiS$_2$)$_{0.07}$ sample also exhibits superiority to other typical thermoelectric systems[30,80,81]. Both dramatically improved hardness and compressive strength can be accounted for by the presence of abundant nanotwins in the matrix. The presence of nanoscale twin boundaries in GeTe effectively provides adequate barriers to block dislocation motion (shown in Supplementary Fig. 20)[31]. Additionally, dislocations can be pinned at the nanoscale twin boundary and react with other dislocations to form a stable, immobile dislocation lock[82]. These factors result in a hardening effect. Therefore, the mechanical properties of hardness and compressive strength for (GeTe)$_{0.93}$(CuBiS$_2$)$_{0.07}$ sample can be both markedly improved. The enhanced mechanical properties extend the potential of GeTe-based materials to serve as a reliable and long-life thermoelectric device. In summary, we demonstrated that high thermoelectric performance and mechanical properties are simultaneously achieved in CuBiS$_2$ alloyed GeTe through nanotwin architecture and inducing ultra-high valley degeneracy. Microstructural observations reveal that dense nanotwins and ordered vacancy arrays are present in the GeTe matrix. The architecture of dense nanotwins, along with ordered vacancy arrays and point defects, leads to significantly reduced lattice thermal conductivity. Furthermore, CuBiS$_2$ alloying resulted in refined valence band alignment, which increases the $N_v$ to an ultra-high value of 22 and produces a sharp increase in power factor over the whole temperature range. Consequently, synergistically optimized electron and phonon transport properties contribute to a high peak ZT value of 2.5 and a record-high average ZT of 1.9 in the (GeTe)$_{0.93}$(CuBiS$_2$)$_{0.07}$ sample. Moreover, hardness and compressive strength are greatly enhanced due to the block of dislocation slip by dense nanotwins, which is beneficial for the assembly and application of thermoelectric devices.

## Methods

### Sample Synthesis

Polycrystalline (GeTe)$_{1-x}$(CuBiS$_2$)$_x$ samples are synthesized by the vacuum melting method. High-purity raw materials of Ge (99.99%), Cu (99.99%), Bi (99.99%), Te (99.99%), and S (99%) powders are weighed according to the nominal compositions. The raw powders were

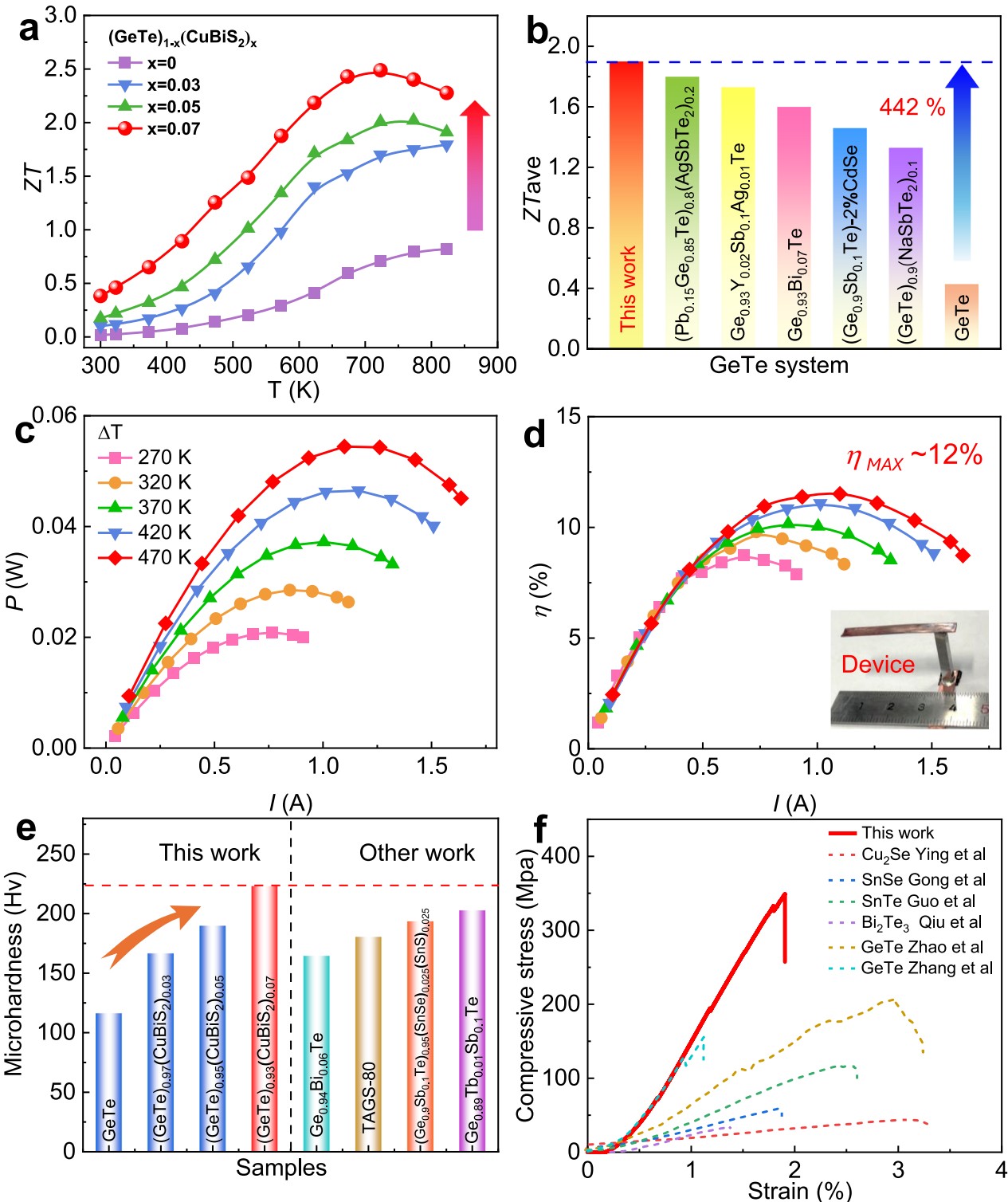

**Fig. 7 | Dimensionless figure of merit *ZT*, conversion efficiency and mechanical properties. a** Temperature-dependent *ZT* values of (GeTe)$_{1-x}$(CuBiS$_2$)$_x$ samples. **b** Comparing the average *ZT* (*ZT*$_{ave}$) in this study with those reported in other works. **c** Current (*I*) dependent output power (*P*), **d** energy conversion efficiency ($\eta$) of a single leg under various temperature differences ($\Delta T$). **e** The Vickers microhardness $H_v$ of (GeTe)$_{1-x}$(CuBiS$_2$)$_x$ samples and comparison with data from literature. **f** The compressive strain-stress and comparison with some typical TE materials.

thoroughly blended before loading into quartz tubes. These tubes were evacuated to high vacuum and hermetically sealed using a high-temperature torch. Subsequent thermal treatment involved heating the samples in a muffle furnace, holding them isothermally at 1273 K for 8 hours, followed by rapid ice water quenching. The samples achieved then underwent a 3-day annealing process at 923K. The

resulting ingots were manually pulverized using a mortar and pestle to obtain fine powders. These powders were loaded into 13-mm-diameter graphite dies and sintered into high relative density cylindrical samples under a uniaxial pressure of 50 MPa using a LABOX-110h system in vacuum. These yielded samples are in dimensions of 13 mm in diameter and 10 mm in height.

## Material characterization

The crystal structure and phase information analysis of the powder samples are supported by (XRD) Bruker D8 Advance instrument with $K\alpha$ radiation ($\lambda$ = 0.154060 nm). DSC analysis was conducted using a Mettler Toledo TGA/DSC3+ instrument at a heating rate of 20 K/min under a nitrogen atmosphere to detect the phase transition temperature. The scanning electron microscopy (SEM) (FEI Quanta 250 FEG) equipped with the energy dispersive spectrometry (EDS) (Inca, Oxford instruments) was performed on the microstructure investigation of the samples. Micrographs were acquired primarily using the Secondary Electron Detector (SED) at an acceleration voltage of 20 kV, a beam condition set to a spot size of 3.0, and a working distance of 10 mm to optimize both image contrast and EDS signal. JEOL JEM-ARM300F2 aberration-corrected scanning transmission electron microscope (STEM) operated at an acceleration voltage of 300 kV. The STEM imaging was conducted with a probe size of approximately 8c, a convergence semi-angle of about 25 mrad, and collection angles ranging from 90 to 370 mrad. Samples for STEM observation were prepared by mechanical grinding followed by ion milling (GATAN 691). The thinning procedure consisted of two stages: an initial coarse thinning at a higher voltage (4–5 kV), followed by a final precision polishing at a lower voltage (1–2 kV) to obtain an electron-transparent area free. Both Seebeck coefficient and electrical conductivity were characterized using an Ulvac-Riko ZEM-3 system under a helium atmosphere from 300–823 K. The total thermal conductivity $\kappa$ was calculated from the equation: $\kappa = \lambda d C_p$. Thermal diffusivity was measured directly via the laser flash technique on a Netzsch LFA457 instrument. The specific heat capacity ($C_p$) was calculated used Dulong-Petit limit (Supplementary Fig. 21). The density ($d$) of samples was determined by a density meter using the Archimedes method. The uncertainties for the electrical conductivity and Seebeck coefficient measurements are both 5%. The uncertainty for the total thermal conductivity is about 12% (comprising uncertainties of 5% for the thermal diffusivity, 5% for the specific heat, and 2% for the density). The combined uncertainty for all measurements involved in the calculation of the ZT is approximately 20%. The Hall carrier concentration ($n$) and carrier mobility ($\mu$) were calculated from the formulas $n = 1/eR_H$ and $\mu = R_H/\rho$, respectively. The Hall coefficients ($R_H$) were measured by the van der Pauw method on the Hall-effect measurement system (Ecopia HMS-3000) at room temperature. Vickers hardness measurements were performed using an HVS-1000 instrument, applying a load of 1 N for a dwell time of 10 seconds to ensure measurement accuracy. Compressive strength was tested on an Instron 7648 machine with a loading rate of 0.05 mm/min.

## Theoretical calculations

We use the projected augmented wave (PAW)[83] scheme with the generalized gradient approximation of Perdew-Burke-Ernzerhof (PBE)[84] for the electronic exchange-correlation functional in the Vienna ab initio simulation package (VASP). A cutoff energy of 450 eV is applied for the plane wave expansion, and the Monkhorst-Pack[85] approach is used to sample the Brillouin zones with a roughly constant density of k-points (30 Å$^3$). The structures are fully optimized until the maximum force on each atom is below 0.01 eV. To simulate the experimentally suggested GeTe alloying with CuBiS$_2$, we perform the special quasi-random structures (SQS)[86,87] method in the Alloy Theoretic Automated Toolkit (ATAT) code. This method can be used to construct the solid solution structure or randomly occupy the lattice during doping. Under this approximation, 8% Cu + 8% Bi and 16% S are randomly doped at the Ge and Te sites in the cubic GeTe phase (Ge$_{63}$Cu$_6$Bi$_6$Te$_{63}$S$_{12}$, Supplementary Fig. 7b). To compare the band structures with or without dopants, the band unfolding method (the BandUP code)[88] is used to unfold the SQS supercell band structure into the primitive Brillouin zone of the pristine GeTe compound.

## Thermoelectric device fabrication and characterization

The thermoelectric single-leg module was fabricated by first sintering the material via SPS, then cutting it into a block measuring 2.95 × 3.07 × 14.2 mm$^3$, and finally soldering it between two copper plates using a commercial Sn$_{64}$Bi$_{35}$Ag$_1$ solder. The soldering process was conducted in a glove box, where a hot plate and a heater block were employed to simultaneously heat both ends of the leg. Subsequently, the device's electrical power output and conversion efficiency were measured using a home-built test system. More details were given in previously reported work[22].

## Data availability

The authors declare that all data supporting the findings of this study are available within the article and its Supplementary Information files or from the corresponding author.

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

## Acknowledgements

The work was supported by the National Natural Science Foundation of China (No. 52071182 to G.D.T., 52472250 to H.J.W., and 12474016 to Y.S.Z.), "Qinglan Project" of the Young and Middle-aged Academic Leader of Jiangsu Province (to G.D.T.), the Fundamental Research Funds for the Central Universities (No. 202510 to G.D.T.), and the program of "Distinguished Expert of Taishan Scholar" (No. tstp20221124 to Y.S.Z.).

## Author contributions

G.D.T. conceived the idea, designed the experiments, and supervised the research. S.L. prepared samples, analyzed data, and wrote the paper. Y.X.Y., Y.Z., and H.J.W. accomplished the microstructural characterizations and analyzed data. X.Y.F., X.B.L., and Y.S.Z. carried out the DFT calculations. Y.G. performed the fabrication and measurements for the single-leg module. J.J.N., P.B.P., and G.Z.L. helped measure the properties. All authors analyzed the results and coedited the manuscript.

## Competing interests

The authors declare no competing interests.
