## [Transparent Peer Review file · Nature Communications]

Nanotwin Architecture and Ultra-high Valley Degeneracy Lead to High Thermoelectric Performance in GeTe-based Thermoelectric Materials

Corresponding Author: Professor Guodong Tang

Version 0:

Reviewer comments:

Reviewer #1

(Remarks to the Author)

Li et al. reports CuBiS₂ alloyed GeTe-based Thermoelectric Materials with a high peak ZT of 2.5 as well as an exceptional average ZT of 1.9 through nanotwin architecture and inducing ultra-high valley degeneracy. DFT calculations reveal that CuBiS₂ alloying realizes refined valence band alignment in GeTe, generating an ultra-high valley degeneracy of 22. Although the reported zT values are comparable to typical GeTe performance, the mechanism by which CuBiS₂ alloying induces valence-band alignment needs further validation. Given the chemical incompatibility between CuBiS₂ and GeTe, its realistic solubility and incorporation into the GeTe matrix remains uncertain. Moreover, several issues relating to claims as enlisted below require major revision.

1. How do chemical incompatibility issues of incorporating CuBiS₂ (a sulfide compound) into GeTe (telluride matrix), by alloying affect the overall phase stability? Moreover, discussion relating to solubility and its limits needs further clarification.
2. The observed increase in lattice parameter (Figure 2d) requires a more rigorous explanation, including how Cu, Bi, and S atoms occupy lattice sites in GeTe, especially given the possibility of structural phase transitions. Furthermore, the phase constitution and quantitative phase fractions present in the alloyed samples are not adequately established/clarified.
3. With valley degeneracy as high as 22, very large effective mass is likely. How does it vary in comparison to previous reports and theoretical evaluation? Moreover, the physical origin and its compatibility with the expected band symmetry require more detailed justification.
4. Figure 6(e) indicates increasing power factor with increasing CuBiS₂ content. Why is higher content not explored?
5. Discuss thermal stability and temperature dependent heat capacity of the samples. Particularly, how does CuBiS₂ alloying and nanotwin architecture affect the specific heat capacity of the GeTe.?
6. A comparative analysis with respect to published studies on GeTe may be included to better comprehend the relevance of ultra-high valley degeneracy.
7. To what extent does the observed enhancement in zT translate into an actual improvement in overall energy conversion efficiency? Presenting device-level efficiency measurements would strengthen and substantiate the claims of high thermoelectric performance.
8. Comment on the measurement error. Similarly, discuss the propagation errors and their impact on the accuracy of the ZT measurements for the synthesized alloys.
9. There are prevailing typos and errors which need to be thoroughly checked and corrected.

Reviewer #2

(Remarks to the Author)

Dear Editor,

I have reviewed the manuscript "Nanotwin Architecture and Ultra-high Valley Degeneracy Lead to High Thermoelectric Performance in GeTe-based Thermoelectric Materials" and recommend its publication after revision. Below I provide my detailed comments for the editor and the authors.

What are the noteworthy results?

The authors demonstrate experimentally that alloying GeTe with Cu, Bi, and S enhances the power factor and reduces the lattice thermal conductivity, increasing the average zT over the relevant temperature range from 0.4 to 1.9. They attribute these improvements to a combination of nanostructuring (twin boundaries, vacancy arrays, and point defects generated by alloying), which lowers the lattice thermal conductivity, together with band structure optimization induced by alloying, which enhances the power factor.

Is the work significant?

This work combines several strategies to enhance thermoelectric performance, provides a reasonable explanation of the observed trends, and reaches a high average zT . The approach is relevant for the design of high-efficiency modules based on GeTe, and similar strategies could be transferred to other thermoelectric materials.

Does the work support the conclusions and claims, or is additional evidence needed?

In general, the conclusions are well supported by the data. There are, however, a few points that should be clarified or further discussed:

How were the point defect, grain boundary, and vacancy densities used in the Debye–Callaway model determined or estimated? Please specify the basis for these parameters and their uncertainty.

Are the properties stable upon thermal cycling? Is there microstructure evolution when the samples are heated to the operating temperature (up to 800 K)? Defect densities and distributions might change when heat is applied. Adding information about that can help strengthen the article (e.g. a figure showing several heating–cooling cycles would help the reader understand whether this is an important effect).

In Figure 4e, f, i, and j the authors use HAADF contrast to comment on the distribution of alloying elements and vacancies. Not much can be seen in Figure 4e and f, and based on the distributions shown in Figure 4i and j it is difficult to conclude that Bi, Cu, and S substitute preferentially on specific lattice sites or to localize individual vacancies. I suggest softening these statements or supporting them with additional evidence (for example EDS/EELS mapping or simulations).

Is the methodology sound?

The overall methodology appears sound, but some important experimental details are missing and should be provided to ensure reproducibility:

For SEM: specify detector type (SE or BSE), acceleration voltage, probe current (or beam conditions), and working distance.

For STEM: specify acceleration voltage, convergence semi-angle, collection angles for the HAADF and BF/ADF detectors, and the approximate probe size. Please also describe how the STEM samples were prepared (FIB, ion milling conditions, etc.).

EDS is mentioned in both SEM and STEM, but no EDS data are shown. Either include representative EDS results or clarify how EDS was used and why the data are not shown.

Small comments

Line 50: duplicated “potential”.

Line 80: “secondary phases”.

Line 106: I suggest adding a brief comparison of the obtained average zT values with state-of-the-art GeTe-based thermoelectrics.

Line 163: “magnified” instead of “further amplified”.

Line 171 and 180: Figure 3 is referenced incorrectly.

Line 190: likely also references a wrong figure, please check.

Line 218: Figure 3 is introduced after Figure 4; consider reordering or adjusting the text.

Best regards,

Version 1:

Reviewer comments:

Reviewer #1

(Remarks to the Author)

Authors have addressed the raised issues and considerations.
The manuscript is recommended for acceptance.

Reviewer #2

(Remarks to the Author)

Thank you for your careful revisions. The authors have addressed the requested changes, and the manuscript has improved accordingly. One minor point: I suggest tempering the statement that negligible TGA mass loss implies unchanged defect densities, since TGA mainly indicates mass stability. With this small clarification, I recommend publication.

Reviewer #1

Comment: Li et al. reports CuBiS₂ alloyed GeTe-based Thermoelectric Materials with a high peak ZT of 2.5 as well as an exceptional average ZT of 1.9 through nanotwin architecture and inducing ultra-high valley degeneracy. DFT calculations reveal that CuBiS₂ alloying realizes refined valence band alignment in GeTe, generating an ultra-high valley degeneracy of 22. Although the reported zT values are comparable to typical GeTe performance, the mechanism by which CuBiS₂ alloying induces valence-band alignment needs further validation. Given the chemical incompatibility between CuBiS₂ and GeTe, its realistic solubility and incorporation into the GeTe matrix remains uncertain. Moreover, several issues relating to claims as enlisted below require major revision.

Response:

Thank you for your careful review and the insightful suggestions offered to improve our manuscript. We have provided a point-by-point response to your comments below. We hope the revisions made to the manuscript adequately address your concerns and significantly improve the quality of this work.

Comment 1: How do chemical incompatibility issues of incorporating CuBiS₂ (a sulfide compound) into GeTe (telluride matrix), by alloying affect the overall phase stability? Moreover, discussion relating to solubility and its limits needs further clarification.

Response:

Thanks for your comments. Except for a small amount of Ge precipitates, no obvious impurity phase can be detected in the XRD pattern of (GeTe)_{1-x}(CuBiS₂)_x samples (Fig. R1) as CuBiS₂ alloying content $x \leq 0.07$. This indicates that the CuBiS₂ can be well dissolved into the GeTe matrix. In order to further investigate the distribution of Cu, Bi, and S elements within the GeTe matrix, we performed SEM and corresponding EDS analysis in Fig. R2. It is clearly found that the Cu, Bi, and S elements are homogeneous distribution in the GeTe matrix. This indicates good chemical compatibility of CuBiS₂ with GeTe.

In order to investigate solid solubility of CuBiS₂, we prepared the sample with higher CuBiS₂ alloying content of $x = 0.09$ during revision. In the XRD pattern of (GeTe)_{0.91}(CuBiS₂)_{0.09} sample (Fig. R1), distinct diffraction peaks indexed to the secondary phase of CuBiS₂ were observed. This suggests that the solubility limit of CuBiS₂ in GeTe is lower than 9%.

Fig. R1. X-ray diffraction patterns of $(\text{GeTe})_{1-x}(\text{CuBiS}_2)_x$ powders at room temperature.

Fig. R2. SEM images of the polished surface of the $(\text{GeTe})_{0.93}(\text{CuBiS}_2)_{0.07}$ sample and corresponding Elemental mapping.

Revisions: We have added the discussion on page 6.

"As the CuBiS_2 alloying content reaches $x = 0.09$, distinct diffraction peaks indexed to the secondary phase of CuBiS_2 were observed. This suggests that the solubility limit of CuBiS_2 in GeTe is lower than 9%."

Comment 2: The observed increase in lattice parameter (Figure 2d) requires a more rigorous explanation, including how Cu, Bi, and S atoms occupy lattice sites in GeTe , especially given the possibility of structural phase transitions. Furthermore, the phase constitution and quantitative phase fractions present in the alloyed samples are not

adequately established/clarified.

Response:

Thank you for your constructive suggestion. The lattice constant a and b increase with increasing CuBiS_2 content, which can be attributed to the fact that the ionic radii of both Bi^{3+} (1.03 Å) and Cu^+ (0.77 Å) are larger than that of Ge^{2+} (0.73 Å). Rietveld refinement details of the XRD pattern of $(\text{GeTe})_{0.93}(\text{CuBiS}_2)_{0.07}$ sample are shown in Fig. R3 and Table R1. The XRD patterns of the $(\text{GeTe})_{0.93}(\text{CuBiS}_2)_{0.07}$ sample can be well fitted by the $r\text{-GeTe}$ structure with a space group of $R\bar{3}m$ (160), similar to the parent GeTe phase. Based on our Rietveld refinement, the Cu and Bi atoms occupy the Ge site (Wyckoff position 2c) with coordinates of (0, 0, 0.2437), while the S occupy Te site (Wyckoff position 2c) with coordinates of (0, 0, 0.7629).

The phase constitution and quantitative phase fractions present in the alloyed samples are analyzed, as shown in Table R2. It indicates good dissolution of CuBiS_2 in GeTe when alloying content is less than or equal to $x = 0.07$.

Fig. R3. Rietveld refinement details of $(\text{GeTe})_{0.93}(\text{CuBiS}_2)_{0.07}$ sample.

Table R1. Rietveld refinement details of the $(\text{GeTe})_{0.93}(\text{CuBiS}_2)_{0.07}$ sample.

Atom	Wyckoff position			Occupancy
	x	y	z	
Ge	0.0000	0.0000	0.2437	0.8856
Cu				0.0596

Bi				0.0548
Te	0.0000	0.0000	0.7629	0.8819
S				0.1181

Table R2. Quantitative phase fractions extracted from the XRD Rietveld refinement for $(\text{GeTe})_{1-x}(\text{CuBiS}_2)_x$ samples.

Composition	Phase fraction (%)		
	GeTe	Ge	CuBiS ₂
GeTe	100	/	/
(GeTe)_{0.97}(CuBiS₂)_{0.03}	100	/	/
(GeTe)_{0.95}(CuBiS₂)_{0.05}	100	/	/
(GeTe)_{0.93}(CuBiS₂)_{0.07}	99.73	0.27	/
(GeTe)_{0.91}(CuBiS₂)_{0.09}	97.85	0.46	1.69

Revisions: We have added Table R1 and Table R2 as Table S1 and Table S2 to the supporting information. Related discussion was included into the revised manuscript of page 6-7.

"Rietveld refined XRD patterns and the refinement details can be found in Fig. 2c-2d and Fig. S1, respectively. Fig. 2d clearly show that the lattice constant a and b increase from 4.16 Å to 4.20 Å with increasing CuBiS₂ alloying content, which can be attributed to the fact that the ionic radii of Bi³⁺ (1.03 Å) and Cu⁺ (0.77 Å) are both larger than that of Ge²⁺ (0.73 Å). The occupied positions of the corrected atoms are exhibited in Table S1. It is found that the Cu and Bi atoms occupy the Ge site (Wyckoff position 2c) with coordinates of (0, 0, 0.2437), while the S occupy Te site (Wyckoff position 2c) with coordinates of (0, 0, 0.7629). Furthermore, the phase constitution and quantitative phase fractions present in the alloyed samples are clarified (Table S2). It proves the good dissolution of CuBiS₂ in GeTe when alloying content is less than or equal to $x = 0.07$. Because the phase transition progress from r -GeTe to c -GeTe is caused by center cation atom shifting along the [111] direction

(Peierls distortion), we can use the lattice angle α to represent the lattice symmetry¹⁰."

Comment 3: With valley degeneracy as high as 22, very large effective mass is likely. How does it vary in comparison to previous reports and theoretical evaluation? Moreover, the physical origin and its compatibility with the expected band symmetry require more detailed justification.

Response:

Thank you for your comment. For the pristine cubic GeTe phase (*c*-GeTe), its valence band maximum (VBM, or VBM1 in our manuscript) is at the L point in the Brillouin zone (BZ). From the shape of the BZ, we can recognize that its valley degeneracy (N_v) is 4. Moreover, the second valence band (VBM2) at the Σ point is ~ 67 meV lower than the VBM, and its $N_v = 12$.¹ Since enhancing the band convergency plays an essential role in boosting the Seebeck coefficient, many works were dedicated to degenerating the two bands of VBM1 (at L) and VBM2 (at Σ) in GeTe. For example, Cd,² Ti,³ Ca,⁴ and Mn⁵ elements doping in GeTe are proved to realize the band convergency between L and Σ . However, the energy difference between light and heavy bands was not effectively eliminated, weakening the contribution of heavy bands. Thus, the actual contributing valley degeneracy ranges from 4 to 16. Then, a refine band alignment was realized in Zn-doped $\text{Ge}_{1-x}\text{Sb}_x\text{Te}$ and V-doped $\text{Ge}_{1-x}\text{Bi}_x\text{Te}$ samples,^{6, 7} band edges between L and Σ nearly locate at the same energy level ($\Delta E^{\text{VBM1-VBM2}} \leq 0.01$ eV), leading to a total band degeneracy of 16 ($N_v^{\text{VBM1}} = 4$, $N_v^{\text{VBM2}} = 12$).

In previous reported works, they focused on promoting band degeneracy of L and Σ . However, from the band structures of *c*-GeTe, we notice that there has the third valence band (VBM3, $N_v=6$) along the Γ -X direction, which is always ignored in former works. If we could reduce the energy difference between third valence band and VBM1, we can further increase the band degeneracy (N_v) and involve more channels for carrier transport. Therefore, by alloying CuBiS₂ in GeTe, we enable the band alignment not only between the VBM1 and VBM2, but also between VBM1 and VBM3 (along the Γ -X). In other words, we successfully achieve the band convergency of three valence bands (VBM1, VBM2 and VBM3), which contribute to a total band degeneracy of 22 ($N_v^{\text{VBM1}} = 4$, $N_v^{\text{VBM2}} = 12$, $N_v^{\text{VBM3}} = 6$). This is a much higher value than those of previous reported literature (Table R3). The theoretically calculated band degeneracy conforms to the band symmetry criteria. The third valence band that realized degenerate after CuBiS₂ alloying is located along the Γ -X k-point

path in the first Brillouin zone, possessing six complete Fermi surface pockets.

Table R3. The Band degeneracy (N_v) realized in typical GeTe-based thermoelectric materials of *c*-GeTe structure.

Material	Contribution	Total degeneracy
c -GeTe ⁸	$N_v^{\text{VBM1}} = 4$ $N_v^{\text{VBM2}} < 12$	< 16
Ge _{0.85} Ca _{0.05} Sb _{0.1} Te ⁴	$N_v^{\text{VBM1}} = 4$ $N_v^{\text{VBM2}} < 12$	< 16
Ge _{0.88} Cd _{0.05} Bi _{0.07} Te ²	$N_v^{\text{VBM1}} = 4$ $N_v^{\text{VBM2}} < 12$	< 16
Ge _{0.86} Mn _{0.10} Sb _{0.04} Te ⁵	$N_v^{\text{VBM1}} = 4$ $N_v^{\text{VBM2}} < 12$	< 16
Ge _{0.86} Sb _{0.1} Zn _{0.04} Te ⁷	$N_v^{\text{VBM1}} = 4$ $N_v^{\text{VBM2}} = 12$	16
Ge _{0.9} V _{0.02} Bi _{0.08} Te ⁶	$N_v^{\text{VBM1}} = 4$ $N_v^{\text{VBM2}} = 12$	16
This work	$N_v^{\text{VBM1}} = 4$ $N_v^{\text{VBM2}} = 12$ $N_v^{\text{VBM3}} = 6$	22

Revision: Table R3 was included into the supporting information (Table S6). Related discussion was included into the revised manuscript.

"In previous reported works, they focused on promoting band degeneracy of L and Σ . Cd¹⁷, Ti⁵⁰, Ca¹⁹, and Mn⁵¹ elements doping in GeTe are proved to realize the band convergency between L and Σ . However, the energy difference between light and heavy bands was not effectively eliminated, weakening the contribution of heavy bands. Thus, the actual contributing valley degeneracy ranges from 4 to 16. Then, a refine band alignment was realized in Zn-doped Ge_{1-x}Sb_xTe and V-doped Ge_{1-x}Bi_xTe samples^{18, 42}, band edges between Σ and L nearly locate at the same energy level ($\Delta E^{\text{VBM1-VBM2}} \leq 0.01$ eV), leading to a total band degeneracy of 16 ($N_v^{\text{VBM1}} = 4$, $N_v^{\text{VBM2}} = 12$). However, from the band structures of *c*-GeTe, we notice that there has the third valence band (VBM3, $N_v=6$) along the Γ -X direction, which is always ignored in former works. If we could reduce the energy difference between third valence band and VBM1, we can further increase the band degeneracy (N_v) and involve more channels for carrier transport. Therefore, by alloying CuBiS₂ in GeTe, we enable the band alignment not only between the VBM1 and VBM2, but also between VBM1 and VBM3, which contribute to a total band degeneracy of 22 ($N_v^{\text{VBM1}} = 4$, $N_v^{\text{VBM2}} = 12$, $N_v^{\text{VBM3}} = 6$). This is a much higher value than those of previous reported literature^{8, 17-19, 42, 51} (Table S6)."

Comment 4: Figure 6(e) indicates increasing power factor with increasing CuBiS₂ content. Why is higher content not explored?

Response:

Thanks for your suggestions. GeTe with higher CuBiS₂ alloying content (x=0.09) was prepared during revision. We measured electrical transport properties of the sample with x=0.09, as shown in Fig. R4. Our results reveal that the power factor shows a significant degradation as CuBiS₂ content further increases to x = 0.09. As CuBiS₂ alloying content exceeds the solid solubility limit, CuBiS₂ precipitates presents within the matrix. These precipitates cause strong carrier scattering, resulting in a sharp deterioration of electrical transport performance. We have added Fig. R4 to Fig. S9 in the supporting information. The related discussion was incorporated into revised manuscript

Fig. R4. Temperature-dependent power factor (PF) of $(\text{GeTe})_{1-x}(\text{CuBiS}_2)_x$ samples.

Revisions:

"The power factor shows a significant degradation (Fig. S9) as CuBiS₂ content further increases to $x = 0.09$. As CuBiS₂ alloying content exceeds the solid solubility limit, CuBiS₂ precipitates presents within the matrix. These precipitates cause strong carrier scattering, resulting in a sharp deterioration of electrical transport performance."

Comment 5: Discuss thermal stability and temperature dependent heat capacity of the samples. Particularly, how does CuBiS₂ alloying and nanotwin architecture affect the specific heat capacity of the GeTe.?

Response:

Thanks for your constructive suggestions. We conducted a detailed investigation on thermal stability of $(\text{GeTe})_{1-x}(\text{CuBiS}_2)_x$ during revision. Firstly, thermogravimetric analysis (TGA) was performed to assess the thermal stability of $(\text{GeTe})_{0.93}(\text{CuBiS}_2)_{0.07}$ material (Fig. R5), and the results indicated negligible weight loss over a wide temperature range, confirming that elemental volatilization is negligible during heating. Then, heating and cooling cycle measurements were carried out on high

performance sample. An almost negligibly small hysteresis is found, suggesting that the material maintains its thermoelectric performance after heating-cooling cycles (Fig. R6). Finally, we measured XRD for high performance $(\text{GeTe})_{0.93}(\text{CuBiS}_2)_{0.07}$ sample before and after cycling thermoelectric measurements, as shown in Fig. R7. The sample maintained consistent phase components before and after cycling thermoelectric measurements. These results demonstrate good thermal stability of the CuBiS_2 alloying sample.

Fig. R5. Thermogravimetric analysis for pure GeTe and $(\text{GeTe})_{0.93}(\text{CuBiS}_2)_{0.07}$ sample.

Fig. R6. The heating-cooling cycling measurement of the high thermoelectric performance $(\text{GeTe})_{0.93}(\text{CuBiS}_2)_{0.07}$ sample. (a) Electrical conductivity, (b) Seebeck coefficient, (c) Thermal conductivity, (d) ZT .

Fig. R7. XRD patterns of $(\text{GeTe})_{0.93}(\text{CuBiS}_2)_{0.07}$ sample before and after heating-cooling cycling measurement.

To investigate the effect of CuBiS_2 alloying and nanotwin architecture on the heat capacity (C_p) for the pure GeTe and $(\text{GeTe})_{0.93}(\text{CuBiS}_2)_{0.07}$ sample, we measured

the C_p under different heating rates by DSC (Fig. R8). It is found that the C_p decreases after CuBiS_2 alloying as compared with pure GeTe . The measured heat capacity of $(\text{GeTe})_{0.93}(\text{CuBiS}_2)_{0.07}$ sample is close to Dulong-Petit limit at temperatures not close to that of phase transition. Most literatures reported high performance GeTe thermoelectrics used Dulong-Petit limit for calculating heat capacity.^{7, 9, 10} To have a consistent comparison on thermal conductivity and ZT of GeTe thermoelectrics, it is therefore meaningful to show the transport property with a heat capacity of Dulong Petit limit as well in this work.

Fig. R8. Heat capacity of $(\text{GeTe})_{1-x}(\text{CuBiS}_2)_x$ samples by DSC measurements, indicating the Dulong-Petit limit as an effective approximation.

Revision: We have added Fig. R6, Fig. R7, Fig. R5 and Fig. R8 as Fig. S13, Fig. S14, Fig. S15, and Fig. S21 in the supporting information. The corresponding discussion was included into the revised manuscript.

"Moreover, the material maintains its thermoelectric performance after heating-cooling cycles, demonstrating excellent thermal stability (Fig. S13). XRD measurements (Fig. S14) reveal that the sample maintained consistent phase components before and after cycling thermoelectric measurements. Thermogravimetric analysis (TGA) results (Fig. S15) reveal negligible weight loss after heating, confirming that elemental volatilization is negligible during heating. SEM and EDS analysis proves that the sample did not significantly degrade under high temperatures (Fig. S16)."

Comment 6: A comparative analysis with respect to published studies on GeTe may be included to better comprehend the relevance of ultra-high valley degeneracy.

Response:

Thank you for your comment. Please see our **Response to your Comment 3**. In previous reported works, they focused on promoting band degeneracy between VBM1 (at L) and VBM2 (at Σ). However, from the band structures of *c*-GeTe, we notice that there has the third valence band (VBM3, $N_v=6$) along the Γ -X direction, which is always ignored in former works. By alloying CuBiS₂ in GeTe, we enable the band alignment not only between the VBM1 and VBM2, but also between VBM1 and VBM3 (along the Γ -X). In other words, we successfully achieve the band convergency of three valence bands (VBM1, VBM2 and VBM3), which contribute to a total band degeneracy of 22 ($N_v^{\text{VBM1}} = 4$, $N_v^{\text{VBM2}} = 12$, $N_v^{\text{VBM3}} = 6$). This is a much higher value than those in the former literatures (Table R4).

Comment 7: To what extent does the observed enhancement in *zT* translate into an actual improvement in overall energy conversion efficiency? Presenting device-level efficiency measurements would strengthen and substantiate the claims of high thermoelectric performance.

Response:

We thank the reviewer for raising this insightful point regarding the translation of material *ZT* to device-level efficiency.

We have performed the single leg measurement for the (GeTe)_{0.93}(CuBiS₂)_{0.07} thermoelectric device during revision. The output voltage (*V*), output power *P*, and conversion efficiency (η) as functions of electric current (*I*) are shown in Fig. R9 and Fig. R10c, d. Experimentally, we achieved a high output power of 54 mW under a temperature difference (ΔT) of 470 K. Eventually, the single leg possesses an outstanding energy conversion efficiency of ~12% with a ΔT of 470 K. The maximum efficiency outperforms most of the reported high-performance thermoelectrics, including those of SnSe¹¹, PbTe¹², Half-Heusler¹³, Bi_{0.4}Sb_{1.6}Te_{3.2}¹⁴, PbSe¹⁵ and GeTe systems¹⁶⁻¹⁹ (Fig. R11). The high energy conversion efficiency in (GeTe)_{0.93}(CuBiS₂)_{0.07} device, proving its high thermoelectric performance.

The experimental efficiency is lower than theoretical conversion efficiency. During the efficiency measurement, thermal radiation significantly impacts the measured heat flow Q_c , which is difficult to measure accurately. As a result, the heat flow through the device was overestimated and finally led to a lower actual efficiency. On the other hand, the soldering internal resistance of the single leg is much higher than the theoretical internal resistance due to the absence of a barrier material. The

above reasons make the equipment efficiency lower than the theoretically calculated efficiency.

Fig. R9. Output voltage (U), input heat flow (Q) of the single leg $(\text{GeTe})_{0.93}(\text{CuBiS}_2)_{0.07}$ thermoelectric device as functions of current (I) under various temperature differences (ΔT).

Fig. R10. Dimensionless figure of merit ZT , conversion efficiency and mechanical properties. (a) Temperature-dependent ZT values of $(\text{GeTe})_{1-x}(\text{CuBiS}_2)_x$ samples. (b)

Comparing the average ZT (ZT_{ave}) in this study with those reported in other works. (c) Current (I) dependent output power (P), (d) energy conversion efficiency (η) of a single leg under various temperature differences (ΔT). (e) The Vickers microhardness H_v of $(GeTe)_{1-x}(CuBiS_2)_x$ samples and comparison with data from literature. (f) The compressive strain-stress and comparison with some typical TE materials.

Fig. R11. Comparison of measured thermoelectric conversion efficiencies for a $(GeTe)_{0.93}(CuBiS_2)_{0.07}$ single-leg device in this study and SnSe, PbTe, Half-Heusler, Cu_2Se , $Bi_{0.4}Sb_{1.6}Te_{3.2}$, PbSe, GeTe systems among comparable temperature differences ΔT .

Revision: We have added Fig. R9, Fig. R11 as Fig S18 and Fig S19 to the supporting information. We have added Fig. R10 as Fig. 7 in the manuscript. We have involved the corresponding discussion on page 18.

"A $(GeTe)_{0.93}(CuBiS_2)_{0.07}$ single leg device was fabricated to demonstrate the application potential. The output voltage (V), output power P , and energy conversion efficiency (η) as functions of electric current (I) are shown in Fig. 7c, d and Fig. S18. Experimentally, we achieved a high output power of 54 mW and an outstanding η of $\sim 12\%$ under a temperature difference ΔT of 470 K. The maximum efficiency outperforms most of the reported high-performance thermoelectrics, including those of SnSe⁶⁸, PbTe⁷⁰, Half-Heusler⁷¹, $Bi_{0.4}Sb_{1.6}Te_{3.2}$ ⁷², PbSe⁷³ and GeTe systems^{29, 40, 74, 75} (Fig. S19)."

Comment 8: Comment on the measurement error. Similarly, discuss the propagation errors and their impact on the accuracy of the ZT measurements for the synthesized alloys.

Response:

We express our appreciation for the valuable suggestion provided. The uncertainties for the electrical conductivity and the Seebeck coefficient measurements are both equal to 5%. The uncertainty for the total thermal conductivity is about 12% (comprising uncertainties of 5% for the thermal diffusivity, 5% for the specific heat, and 2% for the density). The combined uncertainty for all measurements involved in the calculation of the ZT is approximately 20%.

Revisions:

"The uncertainties for the electrical conductivity and the Seebeck coefficient measurements are both equal to 5%. The uncertainty for the total thermal conductivity is about 12% (comprising uncertainties of 5% for the thermal diffusivity, 5% for the specific heat, and 2% for the density). The combined uncertainty for all measurements involved in the calculation of the ZT is approximately 20%."

Comment 9: There are prevailing typos and errors which need to be thoroughly checked and corrected.

Response:

Thank you for your suggestion. We have carefully checked the whole content and corrected the typos and errors.

Reviewer #2**Comment:**

Dear Editor,

I have reviewed the manuscript "Nanotwin Architecture and Ultra-high Valley Degeneracy Lead to High Thermoelectric Performance in GeTe-based Thermoelectric Materials" and recommend its publication after revision. Below I provide my detailed comments for the editor and the authors.

What are the noteworthy results?

The authors demonstrate experimentally that alloying GeTe with Cu, Bi, and S enhances the power factor and reduces the lattice thermal conductivity, increasing the average zT over the relevant temperature range from 0.4 to 1.9. They attribute these improvements to a combination of nanostructuring (twin boundaries, vacancy arrays, and point defects generated by alloying), which lowers the lattice thermal conductivity, together with band structure optimization induced by alloying, which enhances the

power factor.

Is the work significant?

This work combines several strategies to enhance thermoelectric performance, provides a reasonable explanation of the observed trends, and reaches a high average zT. The approach is relevant for the design of high-efficiency modules based on GeTe, and similar strategies could be transferred to other thermoelectric materials.

Does the work support the conclusions and claims, or is additional evidence needed?

In general, the conclusions are well supported by the data. There are, however, a few points that should be clarified or further discussed:

Response:

We appreciate your positive and encouraging comments. We have provided a point-by-point response below and have revised the manuscript according to your suggestions.

Comment 1: How were the point defect, grain boundary, and vacancy densities used in the Debye–Callaway model determined or estimated? Please specify the basis for these parameters and their uncertainty.

Response:

Thank you for your constructive suggestions.

For point defect: Due to the alloying of CuBiS₂ with GeTe, we need to consider the introduction of point defects by Cu, Bi, and S atoms. We calculated the contribution of impurity atoms according to the nominal composition of best performance (GeTe)_{0.93}(CuBiS₂)_{0.07} sample.

The parameter Γ ($\Gamma = \Gamma_M + \Gamma_S$) describes the mass and atomic size contrast with the lattice and represents the strength of point defect phonon scatterings, which includes two components, the scattering parameters due to mass fluctuations Γ_M and strain field fluctuations Γ_S .

$$\Gamma_M = x(1-x) \left(\frac{M_i - \bar{M}}{\bar{M}} \right)^2$$
$$\Gamma_S = x(1-x) \varepsilon \left(\frac{a_i - \bar{a}}{\bar{a}} \right)^2$$

\bar{M} is the average atomic mass, \bar{a} is the average atomic radius, ε is the phenomenological parameter.

For grain boundary: The average grain size (l) in the GeTe matrix after SPS sintering was determined to be approximately $\sim 5 \mu\text{m}$ by statistical analysis of the polished surface using scanning electron microscopy (SEM). The average twin

structure (d) size is estimated by transmission electron microscope (TEM) analysis to be ~ 90 nm. The uncertainty is dominated by the statistical distribution of sizes. We estimate 5% uncertainty in l and 5% uncertainty in d due to the limited sampling area in SEM and TEM.

For vacancy densities: Since Ge vacancy arrays have been demonstrated to form through the absence of a layer of Ge atoms, they are categorized as stacking faults to account for their contribution to phonon scattering. The defect density (N_{va}) was estimated through TEM analysis by counting the number of vacancy arrays intersecting per unit length as $\sim 4 \times 10^6 \text{ m}^{-1}$. The main uncertainty arises from limited sampling area in TEM. We estimate 10% relative error in this measurement.

Revision: We have included the above detail into the supporting information on page 4-6.

Comment 2: Are the properties stable upon thermal cycling? Is there microstructure evolution when the samples are heated to the operating temperature (up to 800 K)? Defect densities and distributions might change when heat is applied. Adding information about that can help strengthen the article (e.g. a figure showing several heating–cooling cycles would help the reader understand whether this is an important effect).

Response:

We appreciate the constructive point from the reviewer. Heating and cooling cycle measurements were carried out on high performance sample during the revision. An almost negligibly small hysteresis is found, suggesting that the material maintains its thermoelectric performance after heating-cooling cycles (Fig. R12). In addition, we measured XRD for high performance $(\text{GeTe})_{0.93}(\text{CuBiS}_2)_{0.07}$ sample before and after cyclic thermoelectric measurements, as shown in Fig. R13. The sample maintained consistent phase components before and after cyclic thermoelectric measurements. These results demonstrate good thermal stability of the CuBiS_2 alloying sample.

To investigate microstructure evolution of the optimal sample when the samples are heated to the operating temperature (up to 800 K), we carried out SEM and EDS characterization of the polished surface of the $(\text{GeTe})_{0.93}(\text{CuBiS}_2)_{0.07}$ sample before (Fig. R14a) and after (Fig. R14b) thermoelectric measurements. The results provide compelling evidence that no obvious microstructural changes were observed. Moreover, no noticeable changes in porosity were detected, further validating that the

sample did not significantly degrade under high temperatures. Ge, Te, Cu, Bi, and S elements all exhibit uniform distribution in the corresponding EDS mapping before and after thermoelectric measurements. Overall, the microstructural integrity of the synthesized $(\text{GeTe})_{0.93}(\text{CuBiS}_2)_{0.07}$ material was effectively preserved when the samples are heated to the operating temperature.

To investigate defect densities and distributions might change when heat is applied, thermogravimetric analysis (TGA) was performed to assess the thermal stability of $(\text{GeTe})_{0.93}(\text{CuBiS}_2)_{0.07}$ material (Fig. R15). And the results indicated negligible weight loss over a wide temperature range, indicating the defect densities and distributions remained largely unchanged.

Fig. R12. The heating-cooling cycling measurement of the high thermoelectric performance $(\text{GeTe})_{0.93}(\text{CuBiS}_2)_{0.07}$ sample. (a) Electrical conductivity, (b) Seebeck coefficient, (c) Thermal conductivity, (d) ZT .

Fig. R13. XRD patterns of $(\text{GeTe})_{0.93}(\text{CuBiS}_2)_{0.07}$ sample before and after heating-cooling cycling measurement.

Fig. R14. SEM image and corresponding EDS mappings of the surface for the $(\text{GeTe})_{0.93}(\text{CuBiS}_2)_{0.07}$ sample before (a) and after (b) measurements. The EDS mappings for Cu, Ge, Bi, Te, and S are displayed in panels (a1-a5) and (b1-b5), respectively.

Fig. R15. Thermogravimetric analysis for pure GeTe and $(\text{GeTe})_{0.93}(\text{CuBiS}_2)_{0.07}$ sample.

Revision: We have added related discussion into the revised manuscript.

"Moreover, the material maintains its thermoelectric performance after heating-cooling cycles, demonstrating excellent thermal stability (Fig. S13). XRD measurements (Fig. S14) reveal that the sample maintained consistent phase components before and after cycling thermoelectric measurements. Thermogravimetric analysis (TGA) results (Fig. S15) reveal negligible weight loss after heating, confirming that elemental volatilization is negligible during heating. SEM and EDS analysis proves that the sample did not significantly degrade under high temperatures (Fig. S16)."

Comment 3: In Figure 4e, f, i, and j the authors use HAADF contrast to comment on the distribution of alloying elements and vacancies. Not much can be seen in Figure 4e and f, and based on the distributions shown in Figure 4i and j it is difficult to conclude that Bi, Cu, and S substitute preferentially on specific lattice sites or to localize individual vacancies. I suggest softening these statements or supporting them with additional evidence (for example EDS/EELS mapping or simulations).

Response:

Thank you for your thoughtful comments. We agree that directly and unambiguously determining the exact lattice sites of specific elements (Bi, Cu, S) or individual vacancies based solely on the Z-contrast (atomic number contrast) images in Fig. 4e, f, i, j is indeed challenging.

In response to your suggestion, we have revised the corresponding statements in

the manuscript on page 9. Specifically, the phrase "suggesting that heavily doped Bi atoms and lighter Cu atoms preferentially substitute into the Ge sites" has been modified to state: "the Z-contrast intensity of the Ge atomic columns exhibits significant local variations (inhomogeneity), indicating pronounced compositional fluctuations at the microscopic scale. Such local compositional fluctuations serve as a key microstructural mechanism for introducing mass-field perturbations and enhancing phonon scattering."

Furthermore, to strengthen the logical basis of this revised interpretation, we have added an explanation of the HAADF imaging principle—where intensity is approximately proportional to Z^2 —before the analysis. This logically links the observed inhomogeneity in Ge column contrast to "compositional fluctuations" (which may arise from Ge vacancies and substitution by atoms of different Z). These modifications directly address the core concern: replacing an earlier, potentially overinterpreted conclusion with a more accurate and defensible description of microstructural features (compositional inhomogeneity), while providing the necessary rationale through the Z^2 dependence to make the argument more rigorous without exceeding the support of the evidence.

We believe these revisions have appropriately softened the conclusions, better aligned the discussion with the evidence, and enhanced the clarity and rigor of the presentation. Thank you again for this valuable suggestion.

Revision: We have revised the corresponding statements in the manuscript on page 9. "The intensity in the HAADF-STEM image is approximately proportional to the atomic number Z^2 ."

"the Z-contrast intensity of the Ge atomic columns exhibits significant local variations (inhomogeneity), indicating pronounced compositional fluctuations at the microscopic scale. Such local compositional fluctuations serve as a key microstructural mechanism for introducing mass-field perturbations and enhancing phonon scattering."

Comment 4:

Is the methodology sound?

The overall methodology appears sound, but some important experimental details are missing and should be provided to ensure reproducibility:

For SEM: specify detector type (SE or BSE), acceleration voltage, probe current (or beam conditions), and working distance.

For STEM: specify acceleration voltage, convergence semi-angle, collection angles for the HAADF and BF/ADF detectors, and the approximate probe size. Please also describe how the STEM samples were prepared (FIB, ion milling conditions, etc.).

Response:

Thank you for your comments.

Detailed SEM Parameters: The microstructure and composition analysis were performed using a FEI Quanta 250 FEG scanning electron microscope equipped with an EDS detector (Inca, Oxford instruments). Micrographs were acquired primarily using the Secondary Electron Detector (SED) at an acceleration voltage of 20 kV, a beam condition set to a spot size of 3.0, and a working distance of 10 mm to optimize both image contrast and EDS signal.

Detailed STEM Parameters: The characterization was performed using JEOL JEM-ARM300F2 aberration-corrected scanning transmission electron microscope (STEM) operated at an acceleration voltage of 300 kV. The STEM imaging was conducted with a probe size of approximately 8 μ m, a convergence semi-angle of about 25 mrad, and collection angles ranging from 90 to 370 mrad. Samples for STEM observation were prepared by mechanical grinding followed by ion milling. The ion milling process was carried out using Ar⁺ ions with the specimen mounted on a liquid nitrogen-cooled stage to minimize beam-induced damage. The thinning procedure consisted of two stages: an initial coarse thinning at a higher voltage (4-5 kV), followed by a final precision polishing at a lower voltage (1-2 kV) to obtain an electron-transparent area free of amorphous surface contamination, suitable for high-resolution imaging.

Revision: Experimental details were provided in revised manuscript on Page 21.

"The scanning electron microscopy (SEM) (FEI Quanta 250 FEG) equipped with the energy dispersive spectrometry (EDS) (Inca, Oxford instruments) was performed on the microstructure investigation of the samples. Micrographs were acquired primarily using the Secondary Electron Detector (SED) at an acceleration voltage of 20 kV, a beam condition set to a spot size of 3.0, and a working distance of 10 mm to optimize both image contrast and EDS signal."

"JEOL JEM-ARM300F2 aberration-corrected scanning transmission electron microscope (STEM) operated at an acceleration voltage of 300 kV. The STEM imaging was conducted with a probe size of approximately 8 μ m, a convergence

semi-angle of about 25 mrad, and collection angles ranging from 90 to 370 mrad. Samples for STEM observation were prepared by mechanical grinding followed by ion milling (GATAN 691). The thinning procedure consisted of two stages: an initial coarse thinning at a higher voltage (4-5 kV), followed by a final precision polishing at a lower voltage (1-2 kV) to obtain an electron-transparent area free."

Comment 5: EDS is mentioned in both SEM and STEM, but no EDS data are shown. Either include representative EDS results or clarify how EDS was used and why the data are not shown.

Response:

We are sorry that we made a type error. We only use EDS in SEM measurements. We have provided the SEM-EDS mapping results in the revised supporting information (Fig. S16).

Comment 6: Small comments Line 50: duplicated "potential".

Response: Thanks for your comment. The duplicated "potential" has been deleted.

Comment 7: Line 80: "secondary phases".

Response:

Thanks for your comment. We have corrected the mistake.

Comment 8: Line 106: I suggest adding a brief comparison of the obtained average zT values with state-of-the-art GeTe-based thermoelectrics.

Response:

Thanks for your suggestions. We compare average ZT values with state-of-the-art GeTe-based thermoelectrics in Fig. R10b (Fig. 7b in the main text).

Comment 9: Line 163: "magnified" instead of "further amplified".

Response:

Thanks for your comment. We have modified the words.

Comment 10: Line 171 and 180: Figure 3 is referenced incorrectly.

Response:

Thanks for your comment. We have corrected this error.

Comment 11: Line 190: likely also references a wrong figure, please check.

Response:

Thanks for your comment. We have corrected this error.

Comment 12: Line 218: Figure 3 is introduced after Figure 4; consider reordering or adjusting the text.

Response:

Thank you for comment. In original manuscript, we first discussed the variation trend of thermal conductivity in Fig. 3a and 3b. Then, we explained the origin of the reduction in lattice thermal conductivity according to TEM characterization results from Fig. 4. We further elucidate the underlying mechanism responsible for the decrease in lattice thermal conductivity by Debye-Callaway model calculations in Fig. 3c and 3d.

We hope that the revised manuscript is now suitable for publication in *Nature Communications*. Thank you for your time and consideration.

Sincerely and best regards

Yours sincerely

Guodong Tang

Reference

1. Liu, W. et al. High - Performance GeTe - Based Thermoelectrics: from Materials to Devices. *Adv. Energy Mater.* **10**, 2000367 (2020).
2. Hong, M. et al. Arrays of Planar Vacancies in Superior Thermoelectric $\text{Ge}_{1-x}\text{-yCd}_x\text{Bi}_y\text{Te}$ with Band Convergence. *Adv. Energy Mater.* **8**, 1801837 (2018).

3. Zhu, C. et al. Realizing high thermoelectric performance in GeTe by defect engineering on cation sites. *Journal of Materials Chemistry C* **10**, 9052-9061 (2022).
4. Li, S. et al. Band flattening and localized lattice engineering realized high thermoelectric performance in GeTe. *J. Mater. Chem. A*, (2025).
5. Zheng, Z. et al. Rhombohedral to Cubic Conversion of GeTe via MnTe Alloying Leads to Ultralow Thermal Conductivity, Electronic Band Convergence, and High Thermoelectric Performance. *J. Am. Chem. Soc.* **140**, 2673-2686 (2018).
6. Sun, Q. et al. Versatile Vanadium Doping Induces High Thermoelectric Performance in GeTe via Band Alignment and Structural Modulation. *Adv. Energy Mater.* **11**, 2100544 (2021).
7. Hong, M. et al. Strong Phonon-Phonon Interactions Securing Extraordinary Thermoelectric Ge_{1-x}Sb_xTe with Zn-Alloying-Induced Band Alignment. *J. Am. Chem. Soc.* **141**, 1742-1748 (2019).
8. Hong, M. et al. Thermoelectric GeTe with Diverse Degrees of Freedom Having Secured Superhigh Performance. *Adv. Mater.* **31**, e1807071 (2019).
9. Dong, J. F. et al. Medium-temperature thermoelectric GeTe: vacancy suppression and band structure engineering leading to high performance. *Energy & Environmental Science* **12**, 1396-1403 (2019).
10. Li, J. et al. Low-Symmetry Rhombohedral GeTe Thermoelectrics. *Joule* **2**, 976-987 (2018).
11. Shi, X.-L. et al. A Solvothermal Synthetic Environmental Design for High-Performance SnSe-Based Thermoelectric Materials. *Advanced Energy Materials* **12**, 2200670 (2022).
12. He, H. et al. Multicomponent Synergistic Doping Enables High-efficiency n-Type PbTe Thermoelectric Devices. *Small* **21**, 2408864 (2025).
13. Xing, Y. et al. High-efficiency half-Heusler thermoelectric modules enabled by self-propagating synthesis and topologic structure optimization. *Energy & Environmental Science* **12**, 3390-3399 (2019).
14. Zhuang, H.-L. et al. Thermoelectric Performance Enhancement in BiSbTe Alloy by Microstructure Modulation via Cyclic Spark Plasma Sintering with Liquid Phase. *Advanced Functional Materials* **31**, 2009681 (2021).
15. Wang, S. et al. Realizing high-performance thermoelectric modules through enhancing the power factor via optimizing the carrier mobility in n-type PbSe crystals. *Energy & Environmental Science* **17**, 2588-2597 (2024).
16. Jiang, Y. et al. Evolution of defect structures leading to high ZT in GeTe-based thermoelectric materials. *Nat. Commun.* **13**, 6087 (2022).
17. Yin, L. C. et al. Interstitial Cu: An Effective Strategy for High Carrier Mobility and High Thermoelectric Performance in GeTe. *Adv. Funct. Mater.* **33**, 2301750 (2023).
18. Qi, X. et al. Efficient rhombohedral GeTe thermoelectrics for low-grade heat recovery. *Materials Today Physics* **45**, (2024).
19. Pei, J. et al. Design and Fabrication of Segmented GeTe/(Bi,Sb)₂Te₃

Thermoelectric Module with Enhanced Conversion Efficiency. *Advanced Functional Materials* **33**, (2023).

Dear Editor,

Many thanks for your correspondence concerning the Reviewers' comments on our manuscript (NCOMMS-25-87470A) entitled "**Nanotwin Architecture and Ultra-high Valley Degeneracy Lead to High Thermoelectric Performance in GeTe-based Thermoelectric Materials**". We are grateful for the Reviewers' positive comment for our manuscript.

Reviewer #1

Comment: Authors have addressed the raised issues and considerations. The manuscript is recommended for acceptance.

Response:

We appreciate the positive comments from the reviewer.

Reviewer #2

Comment: Thank you for your careful revisions. The authors have addressed the requested changes, and the manuscript has improved accordingly. One minor point: I suggest tempering the statement that negligible TGA mass loss implies unchanged defect densities, since TGA mainly indicates mass stability. With this small clarification, I recommend publication.

Response:

We appreciate the positive comments from the reviewer. We corrected the discussion of last Response letter during this revision. The manuscript discussion is consistent with this claim.

"To investigate the mass stability when heat is applied, thermogravimetric analysis (TGA) was performed to assess the thermal stability of $(\text{GeTe})_{0.93}(\text{CuBiS}_2)_{0.07}$ material (Fig. R1). And the results indicated negligible weight loss over a wide temperature range, confirming that elemental volatilization is negligible during heating."

Fig. R1. Thermogravimetric analysis for pure GeTe and $(\text{GeTe})_{0.93}(\text{CuBiS}_2)_{0.07}$ sample.

We hope that the revised manuscript meets the standards for publication in *Nature Communications*. Thanks for your time and consideration.

With kind regards,

Guodong Tang

Nanjing University of Science and Technology,

Nanjing 210094, China.

E-mail: tangguodong@njust.edu.cn

Fax: +86-25-84315159; Tel: +86-25-84315159